# Distributional Reinforcement Learning for Multi-Dimensional Reward Functions

**Pushi Zhang**[*][†]
Tsinghua University
zpschang@gmail.com

**Xiaoyu Chen**[*][†]
Tsinghua University
chen-xy21@mails.tsinghua.edu.cn

**Li Zhao**[‡]
Microsoft Research Asia
lizo@microsoft.com

**Wei Xiong**[†]
The Hong Kong University
of Science and Technology
wxiongae@connect.ust.hk

**Tao Qin**
Microsoft Research Asia
taoqin@microsoft.com

**Tie-Yan Liu**
Microsoft Research Asia
tyliu@microsoft.com

## Abstract

A growing trend for value-based reinforcement learning (RL) algorithms is to capture more information than scalar value functions in the value network. One of the most well-known methods in this branch is distributional RL, which models return distribution instead of scalar value. In another line of work, hybrid reward architectures (HRA) in RL have studied to model source-specific value functions for each source of reward, which is also shown to be beneficial in performance. To fully inherit the benefits of distributional RL and hybrid reward architectures, we introduce Multi-Dimensional Distributional DQN (MD3QN), which extends distributional RL to model the joint return distribution from multiple reward sources. As a by-product of joint distribution modeling, MD3QN can capture not only the randomness in returns for each source of reward, but also the rich reward correlation between the randomness of different sources. We prove the convergence for the joint distributional Bellman operator and build our empirical algorithm by minimizing the Maximum Mean Discrepancy between joint return distribution and its Bellman target. In experiments, our method accurately models the joint return distribution in environments with richly correlated reward functions, and outperforms previous RL methods utilizing multi-dimensional reward functions in the control setting.

## 1 Introduction

Making value network capture more information than scalar value functions is a growing trend in value-based reinforcement learning, which helps the agent gain more knowledge about the environment and has great potentials to improve the sample efficiency of RL agents. In the early stage of deep reinforcement learning, DQN (Mnih et al., 2013) uses the scalar output of the neural network to represent value functions. As value-based RL algorithms evolve, distributional RL algorithms start to use neural networks to approximate return distributions for each state-action pair, and take action based on the expectation of return distributions. By capturing the randomness in return as auxiliary tasks, distributional RL agents can gain more knowledge about the environment and learn better representations to avoid state aliasing (Bellemare et al., 2017). Distributional RL algorithms including C51 (Bellemare et al., 2017), QR-DQN (Dabney et al., 2018b), IQN (Dabney et al., 2018a),

---

[*]Equal contribution.
[†]Work done during an internship at Microsoft Research Asia.
[‡]Corresponding author.

35th Conference on Neural Information Processing Systems (NeurIPS 2021).

FQF (Yang et al., 2019) and MMDQN (Nguyen et al., 2020) achieve substantial performance gain compared to DQN.

In another line of work, HRA (Van Seijen et al., 2017) and RD$^2$ (Lin et al., 2020) consider the setting where multiple sources of reward exist in the environment and modify the value network to model source-specific value function for each source of reward. In these works, the outputs of value networks can be interpreted as multiple source-specific value functions, and the agent takes action based on the sum of all source-specific value functions. Similar to return distribution, estimating the source-specific value functions can be seen as auxiliary tasks, which serve as additional supervision for how the total reward is composed and enable the agent to learn better representations. Several previous works support that it is beneficial to model source-specific value functions (Boutilier et al., 1995; Sutton et al., 2011a; Van Seijen et al., 2017; Lin et al., 2020).

Towards providing more supervision signals and enabling agents to gain more knowledge about the environment, we propose to capture the correlated randomness in source-specific returns. Specifically, we consider the source-specific returns from all sources of rewards as a multi-dimensional random variable, and capture its joint distribution to model the randomness of returns from different sources. This provides an informative learning target for our agent. The framework is general and can be extended to capture the correlated randomness of other types of random variables than rewards. For example, we can capture the correlation between achieving different goals in goal-conditioned reinforcement learning (Schaul et al., 2015), or visiting different states in successor representation (Kulkarni et al., 2016). In this paper, we focus on the method for learning joint return distribution of given source-specific rewards and leave the extension to more general settings for future work.

Following existing works on distributional RL, we study the convergence of the Bellman operator and propose an empirical algorithm to approximate the Bellman operator. First, we define the joint distributional Bellman operator, and prove its convergence under the Wasserstein metric. To derive an empirical algorithm, our proposed method (MD3QN) approximates the joint distributional Bellman operator by minimizing the Maximum Mean Discrepancy (MMD) loss over joint return distributions and its Bellman target. MMD holds desirable properties that, it is a metric over joint distribution and its square can be unbiasedly optimized with batch samples. This enables our algorithm to approximate the Bellman operator accurately when the number of samples goes to infinity and the loss is minimized to zero. In experiments on Atari games and other environments with pixel inputs, our method accurately models the multi-dimensional joint distribution from multiple sources of reward. Moreover, our algorithm outperforms previous work HRA which also separately models multiple sources of reward on Atari game environments.

Our contributions can be summarized as follows:

- We propose a distributional RL algorithm, MD3QN, that extends distributional RL algorithms to model the joint return distribution from multiple sources of reward.

- We establish convergence results for the joint distributional Bellman operator, and our proposed algorithm MD3QN approximates this Bellman operator by minimizing MMD loss over joint return distribution and its Bellman target.

- Empirically, our method outperforms previous RL algorithms utilizing multiple sources of reward, and accurately models the joint return distribution from all sources of rewards.

## 2  Background

### 2.1  Notations and Problem Setting

Since our work considers multiple sources of reward exist in the environment, our problem setting is slightly different from the traditional RL setting in reward function. We consider a Markov Decision Process with multiple sources of reward defined by $(\mathcal{S}, \mathcal{A}, P, \boldsymbol{R}, \rho_0, \gamma)$, where $\mathcal{S}$ is the set of states, $\mathcal{A}$ is the finite set of actions, $P : \mathcal{S} \times \mathcal{A} \rightarrow \mathcal{P}(\mathcal{S})$ denotes the transition probability, $\boldsymbol{R} : \mathcal{S} \times \mathcal{A} \rightarrow \mathcal{P}(\mathbb{R}^N)$ denotes the reward function for a total of $N$ sources of reward, $\rho_0 : \mathcal{P}(\mathcal{S})$ denotes the distribution of initial state $S_0$, $\gamma \in (0, 1)$ is the discount factor. Given a policy $\pi : \mathcal{S} \rightarrow \mathcal{P}(\mathcal{A})$, a trajectory is generated by $s_0 \sim \rho_0$, $a_t \sim \pi(s_t)$, $s_{t+1} \sim P(s_t, a_t)$ and $\boldsymbol{r}_t = [r_{t,1}, r_{t,2}, ..., r_{t,N}]^\top \sim \boldsymbol{R}(s_t, a_t)$. We use $r_t = \sum_{n=1}^{N} r_{t,n}$ to denote the total reward received at time $t$. The goal for reinforcement

learning algorithms is to find the optimal policy $\pi$ which maximizes the expected total return from all sources, given by $J(\pi) = \mathbb{E}_\pi[\sum_{t=0}^{\infty} \gamma^t \sum_{n=1}^{N} r_{t,n}]$.

Next we describe value-based reinforcement learning algorithms in a general framework. In DQN, the value network $Q(s, a; \theta)$ captures the scalar value function, where $\theta$ is the parameters of the value network. $\theta_0$ is the initial value of $\theta$. In $i$-th iteration, the training objective for $\theta_i$ is[4]

$$L(\theta_i) = \mathbb{E}_{s_t,a_t,r_t,s_{s+1}}[(Q(s_t, a_t; \theta_i) - y_i)^2], \tag{1}$$

$$\text{where } y_i = r_t + \gamma \max_{a'} Q(s_{t+1}, a'; \theta_{i-1}), \tag{2}$$

and $s_t, a_t, r_t, s_{t+1}$ are sampled from the replay memory.

## 2.2 Distributional RL

In distributional RL algorithms such as C51, IQN and MMD-DQN, $\mu(s, a; \theta)$ captures the return distribution for each state action pair $(s, a)$. In $i$-th iteration, the training objective for $\theta_i$ is

$$L(\theta_i) = \mathbb{E}_{s_t,a_t,\boldsymbol{r}_t,s_{t+1}}[d(\mu(s_t, a_t; \theta_i), \eta_i)], \tag{3}$$

$$\text{where } \eta_i = (f_{r_t,\gamma})_{\#}\, \mu(s_{t+1}, a'; \theta_{i-1}), a' = \arg\max_{a} \mathbb{E}_{Z \sim \mu(s_{t+1}, a; \theta_{i-1})}[Z]. \tag{4}$$

Here $f_{r,\gamma}(x) := r + \gamma \cdot x$. Given a probability distribution $\nu \in \mathcal{P}(\mathbb{R})$, $f_{\#}\nu \in \mathcal{P}(\mathbb{R})$ is the pushforward measure defined by $f_{\#}\nu(A) = \nu(f^{-1}(A))$ for all Borel sets $A \subseteq \mathbb{R}$ and a measurable function $f : \mathbb{R} \to \mathbb{R}$ (Rowland et al., 2018). $d$ is some distributional metric different in each method. In C51 $d$ is KL divergence, in MMDQN $d$ is Maximum Mean Discrepancy, while in QR-DQN and IQN $d$ is Wasserstein metric, which is optimized approximately via quantile regression loss.

## 2.3 Hybrid Reward Architecture

HRA (Van Seijen et al., 2017) proposes the hybrid architecture to separately model the value function for each source of reward. With multiple sources of reward, we use $\boldsymbol{Q} : (\mathbb{R}^N)^{\mathcal{S} \times \mathcal{A}}$ and $Q_n : \mathbb{R}^{\mathcal{S} \times \mathcal{A}}$ to denote the vectored value function and the $n$-th value function respectively.

The loss used by HRA is given by:

$$L(\theta_i) = \mathbb{E}_{s_t,a_t,\boldsymbol{r}_t,s_{t+1}}[\|\boldsymbol{Q}(s_t, a_t; \theta_i) - \boldsymbol{y}_i\|_2^2], \tag{5}$$

$$\text{where } y_{i,n} = r_{t,n} + \gamma Q_n(s_{t+1}, a'; \theta_{i-1}), a' = \arg\max_{a} Q_n(s_{t+1}, a; \theta_{i-1}). \tag{6}$$

RD$^2$ (Lin et al., 2020) uses the similar hybrid architecture, which also learns the value function separately for each source of reward, but with slight differences with HRA in how the next action $a'$ is computed. The loss used by RD$^2$ is given by[5]

$$L(\theta_i) = \mathbb{E}_{s_t,a_t,\boldsymbol{r}_t,s_{t+1}}[\|\boldsymbol{Q}(s_t, a_t; \theta_i) - \boldsymbol{y}_i\|_2^2], \tag{7}$$

$$\text{where } \boldsymbol{y}_i = \boldsymbol{r} + \gamma \boldsymbol{Q}(s_{t+1}, a'; \theta_{i-1}), a' = \arg\max_{a} \sum_{n=1}^{N} Q_n(s_{t+1}, a; \theta_{i-1}). \tag{8}$$

# 3 Distributional Reinforcement Learning for Multi-Dimensional Reward Functions

In this paper, we propose to capture the correlated randomness from multiple sources of reward, forcing the agent to gain more knowledge about the environment and learn better representations. Specifically, we consider the joint distribution of returns from different sources of reward. First, we introduce the joint distributional Bellman operator and establish its convergence. Second, we

---

[4]In practical implementations, the iterative process is approximated by substituting $\theta_{i-1}$ by the parameter of the target network which tracks the exponential moving average of the online network parameters $\theta_i$.

[5]In equation (8), $\boldsymbol{y}_i$ is an N-dimensional vector. We use the bold font to denote that the variable is for the multi-dimensional reward function.

introduce an empirical algorithm, which approximates the joint distributional Bellman operator through stochastic optimization with deep neural networks. Inspired by MMDQN (Nguyen et al., 2020), we introduce a temporal difference loss based on Maximum Mean Discrepancy over joint return distributions. Finally, we present full implementation details of our method, including network architectures and algorithms.

## 3.1 Definitions and Settings

We consider the joint return under policy $\pi$ as a random vector $\boldsymbol{Z}^\pi(s, a)$ composed of $N$ random variables $\boldsymbol{Z}^\pi(s, a) = (Z_1^\pi(s, a), \cdots, Z_N^\pi(s, a))^\top$:

$$\boldsymbol{Z}^\pi(s, a) = \sum_{t=0}^\infty \gamma^t \boldsymbol{r}_t, \tag{9}$$

$$\text{where } s_0 = s, a_0 = a, \boldsymbol{r}_t \sim R(\cdot|s_t, a_t), s_{t+1} \sim P(\cdot|s_t, a_t), a_{t+1} \sim \pi(\cdot|s_{t+1}). \tag{10}$$

Here $Z_i^\pi(s, a)$ denotes the $i$-th dimension of $\boldsymbol{Z}^\pi(s, a)$, which is the random variable of the $i$-th source of discounted return. Here the random variables of discounted return in different sources can be correlated. We denote the distribution of $\boldsymbol{Z}^\pi$ as $\boldsymbol{\mu}^\pi = \text{Law}(\boldsymbol{Z}^\pi)$, and the distribution of $i$-th random variable $Z_i^\pi$ as $\mu_i^\pi = \text{Law}(Z_i^\pi)$, where $\boldsymbol{\mu}^\pi \in \mathcal{P}(\mathbb{R}^N)^{\mathcal{S}\times\mathcal{A}}$ and $\mu_i^\pi \in \mathcal{P}(\mathbb{R})^{\mathcal{S}\times\mathcal{A}}$. We let $\boldsymbol{\mu} \in \mathcal{P}(\mathbb{R}^N)^{\mathcal{S}\times\mathcal{A}}$ to denote arbitrary joint distribution over all state-action pairs, and the goal of our algorithm is to let the joint distribution $\boldsymbol{\mu}$ to be as close to the joint distribution $\boldsymbol{\mu}^\pi$ as possible in policy evaluation setting, i.e. to model the real joint return distribution.

To measure how close two $N$-dimensional joint distributions are, we adopt the Wasserstein metric $W_p$. For two joint distributions $\nu_1, \nu_2 \in \mathcal{P}(\mathbb{R}^N)$, the p-Wasserstein metric $W_p(\nu_1, \nu_2)$ on Euclidean distance $d$ is given by:

$$W_p(\nu_1, \nu_2) = \left( \inf_{f\in\Gamma(\nu_1,\nu_2)} \int_{\mathbb{R}^N \times \mathbb{R}^N} d(x, y)^p f(x, y) d^N x d^N y \right)^{1/p}, \tag{11}$$

where $\Gamma(\nu_1, \nu_2)$ is the collection of all distributions with marginal distributions $\nu_1$ and $\nu_2$ on the first and second $N$ random variables respectively. The Wasserstein distance can be interpreted as the minimum moving distance to move all the mass from distribution $\nu_1$ to distribution $\nu_2$.

We further define the supremum-$p$-Wasserstein distance on $\mathcal{P}(\mathbb{R}^N)^{\mathcal{S}\times\mathcal{A}}$ by

$$\bar{d}_p(\boldsymbol{\mu}_1, \boldsymbol{\mu}_2) := \sup_{s,a} W_p(\boldsymbol{\mu}_1(s, a), \boldsymbol{\mu}_2(s, a)), \tag{12}$$

where $\boldsymbol{\mu}_1, \boldsymbol{\mu}_2 \in \mathcal{P}(\mathbb{R}^N)^{\mathcal{S}\times\mathcal{A}}$.

We will use $\bar{d}_p$ to establish the convergence of the joint distributional Bellman operator.

## 3.2 Convergence of the Joint Distributional Bellman Operator

We define the joint distributional Bellman evaluation operator $\mathcal{T}^\pi$ as

$$\mathcal{T}^\pi \boldsymbol{\mu}(s_t, a_t) \overset{D}{:=} \int_{\mathcal{S}} \int_{\mathcal{A}} \int_{\mathbb{R}^N} (f_{\boldsymbol{r}_t,\gamma})_{\#} \boldsymbol{\mu}(s_{t+1}, a_{t+1}) R(d\boldsymbol{r}_t|s_t, a_t)\pi(da_{t+1}|s_{t+1})P(ds_{t+1}|s_t, a_t), \tag{13}$$

where $f_{\boldsymbol{r},\gamma}(\boldsymbol{x}) = \boldsymbol{r} + \gamma\boldsymbol{x}$, where $\boldsymbol{x}, \boldsymbol{r} \in \mathbb{R}^N$ and $f_{\#}\boldsymbol{\mu}$ is the pushforward measure which is a $N$-dimensional extension of the pushforward measure defined in Rowland et al. (2018). The below theorem shows that $\mathcal{T}^\pi$ is a $\gamma$-contraction operator on $\bar{d}_p$.

**Theorem 1** *For two joint distributions $\boldsymbol{\mu}_1$ and $\boldsymbol{\mu}_2$, we have*

$$\bar{d}_p(\mathcal{T}^\pi \boldsymbol{\mu}_1, \mathcal{T}^\pi \boldsymbol{\mu}_2) \leq \gamma\bar{d}_p(\boldsymbol{\mu}_1, \boldsymbol{\mu}_2). \tag{14}$$

The proof for Theorem 1 is provided in Appendix A.1 which can be briefly described as follows. In Lemma 3, we prove that if the Wasserstein distance of two joint distributions is $C$, then after applying the same linear transformation to these two distributions with a scale factor of $\gamma$, the Wasserstein

distance is at most $\gamma C$. In Lemma 4, we prove that the Wasserstein distance of two joint distributions under the same conditional random variable $A$ is no greater than the maximum Wasserstein distance over all possible conditions $A = a$. By Lemma 3 and Lemma 4, we are able to prove the contraction results in Theorem 1.

Following the result of Theorem 1, we consider the following scenario: initially we have a joint distribution $\boldsymbol{\mu}_0 \in \mathcal{P}(\mathbb{R}^N)^{\mathcal{S} \times \mathcal{A}}$ over all state-action pairs, and by iteratively applying the Bellman evaluation operator, we get $\boldsymbol{\mu}_{i+1} = \mathcal{T}^\pi \boldsymbol{\mu}_i$. According to Banach's fixed point theorem, operator $\mathcal{T}^\pi$ has a unique fixed point, which is $\boldsymbol{\mu}^\pi$ by definition. The distance between $\boldsymbol{\mu}_i$ and $\boldsymbol{\mu}^\pi$ decays as $i$ increases. We have the following corollary:

**Corollary 1** *If $\boldsymbol{\mu}_{i+1} = \mathcal{T}^\pi \boldsymbol{\mu}_i$, then as $i \to \infty$, $\boldsymbol{\mu}_i \to \boldsymbol{\mu}^\pi$.*

We also provide contraction proof on the expectation of the optimality operator. The joint distributional Bellman optimality operator $\mathcal{T}$ is defined as

$$\mathcal{T}\boldsymbol{\mu}(s_t, a_t) \overset{D}{:=} \int_{\mathcal{S}} \int_{\mathbb{R}^N} (f_{\boldsymbol{r}_t, \gamma})_\# \boldsymbol{\mu}(s_{t+1}, a') R(d\boldsymbol{r}_t | s_t, a_t) P(ds_{t+1} | s_t, a_t), \tag{15}$$

$$\text{where } a' \in \arg\max_a \mathbb{E}_{\boldsymbol{Z} \sim \boldsymbol{\mu}(s_{t+1}, a)} \sum_{n=1}^{N} Z_n \text{ and } Z_n \text{ is the } n\text{-th element in } \boldsymbol{Z}. \tag{16}$$

**Theorem 2** *For two joint distributions $\boldsymbol{\mu}_1$ and $\boldsymbol{\mu}_2$, we have*

$$||(\mathbb{E}_{\textstyle\sum})(\mathcal{T}\boldsymbol{\mu}_1) - (\mathbb{E}_{\textstyle\sum})(\mathcal{T}\boldsymbol{\mu}_2)||_\infty \le \gamma ||(\mathbb{E}_{\textstyle\sum})\boldsymbol{\mu}_1 - (\mathbb{E}_{\textstyle\sum})\boldsymbol{\mu}_2||_\infty, \tag{17}$$

where the operator $\mathbb{E}_{\textstyle\sum}$ is defined by $(\mathbb{E}_{\textstyle\sum})\boldsymbol{\mu}(s, a) = \mathbb{E}_{\boldsymbol{Z} \sim \boldsymbol{\mu}(s, a)} \sum_{n=1}^{N} Z_n$ for any $(s, a)$, and $Z_n$ is the $n$-th element in $\boldsymbol{Z}$. The operator $\mathbb{E}_{\textstyle\sum}$ converts the joint distribution over all state-action pairs to the corresponding total expected returns over all state-action pairs.

**Corollary 2** *If $\boldsymbol{\mu}_{i+1} = \mathcal{T}\boldsymbol{\mu}_i$, then as $i \to \infty$, $\mathbb{E}_{\boldsymbol{Z} \sim \boldsymbol{\mu}_i(s, a)} \sum_{n=1}^{N} Z_n \to Q^*(s, a)$ for all $(s, a)$.*

Corollary 2 can be interpreted as follows: as we iteratively apply the joint distributional Bellman optimality operator, the expected total return of the joint distribution $\boldsymbol{\mu}_i$ will converge to optimal value function. We refer the readers to Appendix A.1 for detailed proofs for all the lemmas, theorems, and corollaries.

Next, we establish the empirical algorithm which simulates the iterative process in Corollary 1 and Corollary 2 with a neural network approximating the joint distribution $\boldsymbol{\mu}_i$ for all state-action pairs.

### 3.3 Optimizing MMD as Approximation for Bellman Operator

In practical algorithms, the joint distribution $\boldsymbol{\mu}_i(s, a)$ is represented by deep neural network $\boldsymbol{\mu}(s, a; \theta_i)$ which outputs joint distribution for each state-action pair with parameter $\theta_i$. In the $(i+1)$-th iteration, we can only adapt the parameter $\theta_{i+1}$ of the model, and cannot directly apply the Bellman operator as in the tabular case. Moreover, we only have sampled transitions, rather the environment dynamics model $P$. It is desirable to choose a loss that is compatible with stochastic optimization, as an approximation for the Bellman operator.

We achieve this by minimizing the Maximum Mean Discrepancy (MMD) between joint distribution $\boldsymbol{\mu}_{i+1}$ and $\mathcal{T}\boldsymbol{\mu}_i$, which is defined as the equation below.

$$\text{MMD}^2(p, q; k) = \mathbb{E}_{x, x' \sim p} k(x, x') - 2\mathbb{E}_{x \sim p, y \sim q} k(x, y) + \mathbb{E}_{y, y' \sim q} k(y, y'), \tag{18}$$

where $k(\cdot, \cdot)$ is some kernel function and MMD is defined upon $k$, and each pair of random variable $(x, x'), (x, y), (y, y')$ are independent. In our scenario, we use $\text{MMD}^2(\boldsymbol{\mu}_{i+1}(s, a), \mathcal{T}\boldsymbol{\mu}_i(s, a); k)$ as the temporal difference loss. The reason why we choose MMD as the objective function is that MMD is both a metric on distribution and easy to optimize with sampled transitions. The original paper of MMD (Gretton et al., 2012) proves that MMD is a metric on distribution when kernel $k$ is characteristic, from which we can prove that

$$\boldsymbol{\mu}_{i+1} = \mathcal{T}\boldsymbol{\mu}_i \iff \forall (s, a) \in \mathcal{S} \times \mathcal{A}, \text{MMD}^2(\boldsymbol{\mu}_{i+1}(s, a), \mathcal{T}\boldsymbol{\mu}_i(s, a); k) = 0, \tag{19}$$

when the kernel $k$ is characteristic. If the MMD metric is optimized to zero and the appropriate kernel is selected, $\boldsymbol{\mu}_{i+1}$ is the exact result of applying the joint distributional Bellman operator.

In the algorithmic aspect, $\mathcal{T}\boldsymbol{\mu}_i(s,a)$ serves as the target distribution, $\boldsymbol{\mu}_{i+1}(s,a)$ serves as the prediction from online network, and we use $\text{MMD}^2(\boldsymbol{\mu}_{i+1}(s,a),\mathcal{T}\boldsymbol{\mu}_i(s,a);k)$ as temporal difference loss to match the online prediction and target distribution. Next, we establish the method to optimize $\text{MMD}^2(\boldsymbol{\mu}_{i+1}(s,a),\mathcal{T}\boldsymbol{\mu}_i(s,a);k)$ with transition samples. Note that we use $\boldsymbol{\mu}(s,a;\theta_i)$ to represent $\boldsymbol{\mu}_i(s,a)$, so the temporal difference loss can also be written as $\text{MMD}^2(\boldsymbol{\mu}(s,a;\theta_{i+1}),\mathcal{T}\boldsymbol{\mu}(s,a;\theta_i);k)$. We use equation (18) to estimate the gradient of this squared MMD loss with respect to parameter $\theta_{i+1}$. Specifically, the first term in equation (18) can be unbiasedly estimated by drawing multiple independent samples from the online network $\boldsymbol{\mu}(s,a;\theta_{i+1})$; the second term of equation (18) can be unbiasedly estimated by drawing independent samples from the online network $\boldsymbol{\mu}(s,a;\theta_{i+1})$ and from the target distribution $\mathcal{T}\boldsymbol{\mu}(s,a;\theta_i)$[6]. Batch of samples from the target joint distribution $\mathcal{T}\boldsymbol{\mu}(s,a;\theta_i)$ can be collected from only one transition sample $(s,a,\boldsymbol{r},s')$, and Lemma 6 in Appendix A.1 provides detailed proofs. For the third term in equation (18), it has no gradient with respect to the parameter $\theta_{i+1}$, and we can safely ignore it during optimization. In summary, the gradient of the temporal difference loss $\text{MMD}^2$ can be estimated without bias by transition samples. Algorithm 1 summarizes the above analysis and shows how our method computes the gradient estimates of the temporal difference loss $\text{MMD}^2$.

---

**Algorithm 1** Gradient estimation of $\text{MMD}^2$ loss by transition samples

---

    **Require:** Number of samples $M$, kernel $k$, discount factor $\gamma \in (0,1)$
    **Require:** Joint distribution network $\boldsymbol{\mu}(s,a;\theta)$
    **Input:** Transition sample $(s,a,\boldsymbol{r},s')$
    **Input:** Online network parameter $\theta$, target network parameter $\theta'$
    **Output:** Gradient estimation of MMD with respect to $\theta$
1:  $a' \leftarrow \arg\max_a \frac{1}{M}\sum_{m=1}^{M}\sum_{n=1}^{N}(\boldsymbol{Z}_m)_n$, where $\boldsymbol{Z}_{1:M} \overset{i.i.d.}{\sim} \boldsymbol{\mu}(s',a;\theta')$.
2:  Sample $\boldsymbol{Z}_{1:M} \overset{i.i.d.}{\sim} \boldsymbol{\mu}(s,a;\theta)$ (samples from online network)
3:  Sample $\boldsymbol{Z}_{1:M}^{next} \overset{i.i.d.}{\sim} \boldsymbol{\mu}(s',a';\theta')$
4:  $\boldsymbol{Y_i} \leftarrow \boldsymbol{r} + \gamma\boldsymbol{Z}_i^{next}$, for every $1 \le i \le M$ (samples from target distribution)
5:  $\text{MMD}^2 \leftarrow \sum_{1 \le i \le M}\sum_{1 \le j \le M, j \ne i}[k(\boldsymbol{Z}_i,\boldsymbol{Z}_j) - 2k(\boldsymbol{Z}_i,\boldsymbol{Y}_j) + k(\boldsymbol{Y}_i,\boldsymbol{Y}_j)]$
6:  **return** $\nabla_\theta\text{MMD}^2$

---

### 3.4 Network Architecture and Implementation Details

For a given state-action pair $(s,a)$, we use the value network of DQN (Mnih et al., 2013) to compute the low-dimensional (512d) embedding of state $s$, and replace the final layer of DQN network to output a vector with $M \times N \times |\mathcal{A}|$ dimensions, representing $M$ samples from the joint return distribution from $N$ sources of reward for every action.

We follow the kernel selection in MMDQN (Nguyen et al., 2020) and apply the Gaussian kernel function with mixed bandwidth[7].

$$k(x,x') = \sum_{i=1}^{M} k_{w_i}(x,x'),\tag{20}$$

$$\text{where } k_{w_i}(x,x') = e^{-\frac{\|x-x'\|_2^2}{w_i}}.\tag{21}$$

Each $w_i$ can be seen as the square of bandwidth in a Gaussian kernel. We choose the value of squared bandwidth $w_i$ with diverse ranges, to make sure that all scales of reward don't suffer from gradient vanishing. In Appendix A.3.4 and A.3.5, we test the performance of different network architectures and different bandwidth sets to find the optimal network architecture and bandwidth configurations.

---

[6]We only require each sample from the online network $\boldsymbol{\mu}(s,a;\theta_{i+1})$ to be independent of every sample from the target distribution $\mathcal{T}\boldsymbol{\mu}(s,a;\theta_i)$, and multiple samples from the target distribution $\mathcal{T}\boldsymbol{\mu}(s,a;\theta_i)$ can be dependent of each other.

[7]Gaussian kernel is proved to be characteristic (Gretton et al., 2012), so the MMD distance with Gaussian kernel is a metric. It follows that when we use Gaussian kernels with mixed bandwidth, equation (19) still holds.

## 4  Related Work

Distributional RL algorithms propose to model the entire distribution, rather than the expectation, of the random variable return. C51 (Bellemare et al., 2017) is the first distributional RL algorithm, and establishes the convergence of distributional RL algorithms based on the contraction property of the distributional Bellman operator. To approximate such an operator with sample transitions, different distributional algorithms employ different losses over distribution, along with different parameterization for distribution. C51 uses KL-divergence as loss and categorical distribution as parameterization. QR-DQN (Dabney et al., 2018b) takes quantile regression as surrogate loss for the Wasserstein metric, and approximates a set of quantile values with fixed probabilities. IQN (Dabney et al., 2018a) and FQF (Yang et al., 2019) extend QR-DQN to learn with sampled probabilities and self-adjusted probabilities. The current SOTA method in distributional RL is MMDQN (Nguyen et al., 2020), which takes MMD loss with a non-parametric approach by modeling deterministic samples from return distribution. Our method extends the theoretical work of C51, and the empirical algorithm of MMDQN to multi-dimensional returns. While distributional RL algorithms focus on capturing the randomness in return, our method first proposes to capture the correlation between different randomness from different reward sources.

HRA (Van Seijen et al., 2017) proposes a hybrid architecture to separately model the value functions for different sources of rewards. Their work provides empirical justification that learning with hybrid rewards can improve sample efficiency. HRA is built upon the Horde architecture (Sutton et al., 2011b), which trains a separate general value function (GVF) for each pseudo-reward function. The Horde architecture uses a large number of GVFs to model general knowledge about the environment. Previous works on reward decomposition also demonstrate that it is beneficial to learn with multiple reward functions (Lin et al., 2020). Based on HRA, our method further considers the correlated randomness in hybrid-source returns with a joint distribution model.

Bellman GAN (Freirich et al., 2019) shows that distributional RL can be used for multi-dimensional rewards. Compared to this work, we additionally provide the theoretical convergence results for the joint distributional Bellman operator, an algorithm where the gradient of objective function (MMD) can be unbiasedly estimated, and experimental results that directly compare the modeled joint distribution to the true distribution, demonstrating the accuracy of our method under diverse reward correlations. DRDRL (Lin et al., 2019) focuses on decomposing multiple types of rewards from one source reward, while in our paper, we are directly provided with multiple types of rewards as inputs, and model the joint distribution of different sources of rewards.

By modeling the joint return distribution, we provide an informative target to learn, which can be seen as auxiliary tasks. Previous works in RL have constructed various auxiliary tasks to learn better representations, such as methods based on temporal structures (Aytar et al., 2018) and local spatial structures (Anand et al., 2019). Compared with these methods, our method does not entirely focus on learning better state representations. The learned joint distribution is also beneficial to risk-sensitive tasks (Zhang et al., 2020). Besides, our method for learning the joint distribution of multi-dimensional random variables is general, and can be further combined with goal-conditioned RL (Schaul et al., 2015) or successor representation (Kulkarni et al., 2016) to capture correlated randomness in achieving different goals or visiting different states.

## 5  Experimental Results

In our experiments[8], we provide empirical results to answer the following questions:

- On policy evaluation settings, can MD3QN accurately model the joint distribution of multiple sources of reward?
- On policy optimization settings, can MD3QN learn a better policy compared to HRA and distributional RL algorithm on environments with multiple sources of reward?

To answer the first question, we design a maze environment with a rich way to generate correlated sources of rewards, and compare the joint distribution predicted by our method with the true joint distribution $\mu^\pi$. The detailed setting of the environment and results is shown in Section 5.1.

---

[8]Our code for the experiments is available in the supplementary material.

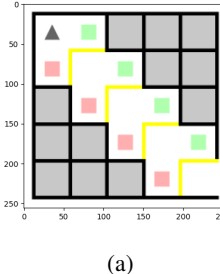 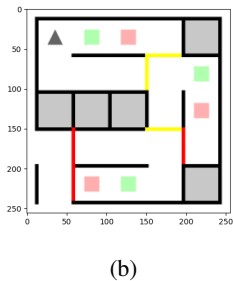 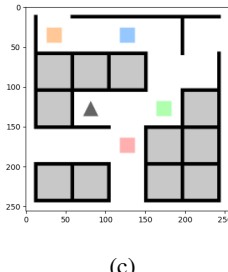

| (a) | (b) | (c) |

Figure 1: Observation of initial states in maze environments. (a): maze environment "maze-exclusive" with two exclusive sources of rewards. (b): maze environment "maze-identical" with two positively correlated sources of rewards. (c): maze environment "maze-multireward" with four correlated sources of rewards.

To answer the second question, we use several Atari games with multiple sources of reward, and provide training curve results of MD3QN compared to previous work HRA which deals with multiple sources of reward. The detailed setting of the environment and results is shown in Section 5.2.

## 5.1 Modeling Joint Return Distribution for Multi-Dimensional Reward Functions

In this experiment, we validate the capacity of MD3QN to model the joint return distribution for multiple sources of rewards on the policy evaluation setting with pixel inputs. The experiment is performed on the maze environments with rich reward correlation signals described as follows.

Figure 1 illustrates three maze environments with different reward correlation properties. The agent (represented by the triangle) is initially located at a specific position in the maze, and the policy $\pi$ is a fixed policy to uniformly choose one direction at random and try to move. If the agent is blocked by the walls in that direction[9], or the agent is trying to return to a previous position, the move has no effect, otherwise, the agent moves in that direction for one block. Each color of the square in the maze represents one source of reward. When the agent reaches the position with reward, it receives a reward of the source aligned with the color of the square[10]. The rules described above are general enough to enable the design of diversely correlated sources of rewards detailed below.

Figure 1(a) shows a maze environment with two exclusive sources of rewards. Once the agent obtains a reward from one side, it cannot obtain the reward from the other side. Figure 1(b) shows a maze environment with two positively correlated sources of rewards, where the agent either gets no reward or gets both of the red and green rewards. Figure 1(c) shows a maze environment with four correlated sources of rewards. In these three environments, the agent needs to capture the negative correlation, positive correlation, and the complex correlation in multi-dimensional rewards respectively to precisely model the joint return distribution.

On each of the three mazes, we train the MD3QN agent for 5 iterations (1.25M frames) to model the joint distribution of all sources of reward, and compare the prediction of joint distribution by the model to the samples from the true distribution $\mu^\pi$ on the initial state of the maze. The results are shown in Figure 2, where each point in the figure is a sample representing one possibility of future discounted return in each source of reward. The result shows that MD3QN accurately models the joint return distribution from different sources of reward.

In Appendix A.3.1, we visualize the modeled joint distribution throughout the training process, where the joint distribution modeled by MD3QN gradually match the true distribution.

It is also worth noting that the observation for the agent is based on pixels, computed by downsampling the image in Figure 1 to the size of $84 \times 84$, which requires the agent to extract information on high-dimensional inputs.

---

[9]Colored walls can only be passed in one way, where yellow walls can be passed from the top or from the left, and red walls can be passed from the bottom or from the right.

[10]More detailed description of how reward is set in these three environments is provided in Appendix A.2.

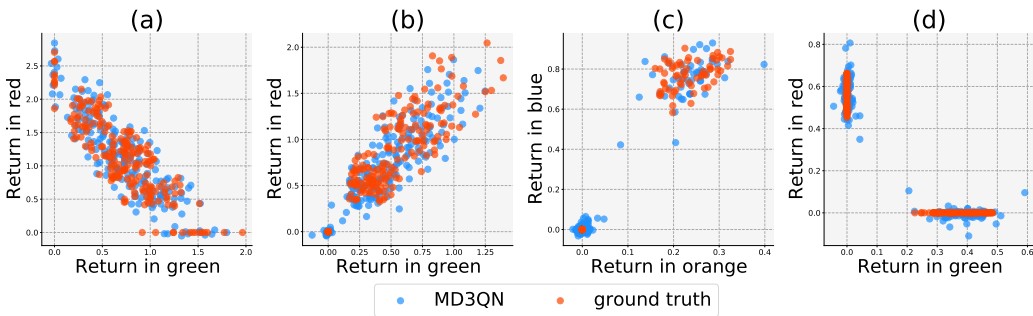

Figure 2: Comparison between samples from joint distribution by MD3QN and samples from the true distribution $\mu^\pi$ in three maze environments. The number of samples is set to be 200. (a): the result of "maze-exclusive" environment. (b): the result of "maze-identical" environment. (c): the result of "maze-multireward" environment of orange and blue reward sources. (d): the result of "maze-multireward" environment of green and red reward sources.

## 5.2 Performance on Atari Games

In this experiment, we compare MD3QN algorithm with Hybrid Reward Architecture (HRA) (Van Seijen et al., 2017) and MMDQN (Nguyen et al., 2020) on Atari games from Arcade Learning Environment (Bellemare et al., 2013). Our implementation of MD3QN is built upon the Dopamine framework (Castro et al., 2018). The hyper-parameter settings used by MD3QN and the environment settings are detailed in Appendix A.2.

We use six Atari games with multiple sources of rewards: AirRaid, Asteroids, Gopher, MsPacman, UpNDown, and Pong. In all of these six games, the primitive rewards can be decomposed into multiple sources of rewards as follows:

- In AirRaid and Asteroids, the agent gets different value of reward for killing different types of monsters.
- In Gopher, the agent gets +80 reward for killing a monster and gets +20 reward for removing holes on the ground.
- In MsPacman, the agent gets {+200, +400, +800, +1600} reward for killing a monster and gets +10 reward for eating beans.
- In UpNDown, the agent gets +400 reward for killing an enemy car, +100 reward for reaching a flag, and +10 reward for being alive.
- In Pong, the agent gets +1 reward for winning a round, and gets -1 reward for losing a round.

We decompose the primitive rewards into multi-dimensional rewards according to the above reward structure, and meanwhile keeping the total reward not changed. The details for the reward decomposition method is provided in Appendix A.2. The training curves of MD3QN compared to HRA (Van Seijen et al., 2017) and MMDQN (Nguyen et al., 2020) are shown in Figure 3.

Figure 3 shows that MD3QN does not outperform MMDQN in performance under several Atari games. We suspect the reason why MD3QN cannot outperform MMDQN is that the performance of Atari games might not be related to the joint distribution. We look forward to further studies on how the additional information captured in distributional RL algorithms helps to improve the final performance, to better interpret the experimental results. Despite that, with the distributional information, MD3QN outperforms or matches the performance of HRA under all six games.

On Atari games, we also show the joint distributions modeled by MD3QN, which is detailed in Appendix A.3.2. Overall, both the reward correlation and the reward scale in the return distribution are reasonably captured by MD3QN. The joint distribution captured by MD3QN show different and reasonable reward correlation across different Atari games.

To show the necessity of modeling the joint distribution, we add another experiment in Appendix A.3.3 showing that only modeling marginal distributions may fail in environments with multiple correlated constraints, from theory and experiment results.

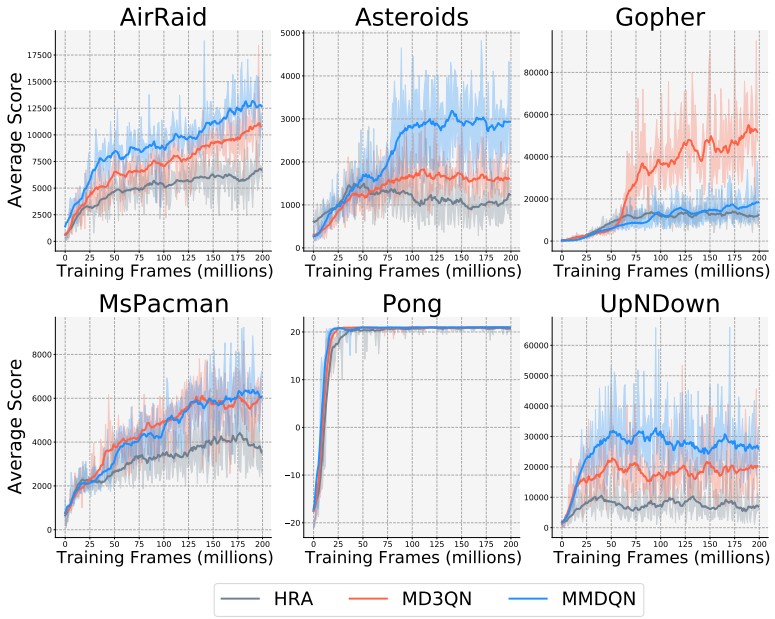

Figure 3: Performance of MD3QN on Atari games compared to HRA and MMDQN. The shaded areas show the standard deviation across 3 seeds.

## 6 Conclusions

In this work, we have proposed Multi-Dimensional Distributional DQN (MD3QN), a distributional RL method that learns a multi-dimensional joint return distribution for multiple sources of rewards. The effectiveness of our method is verified on pixel-input environments in terms of both the quality of modeled joint distribution, and the final performance of learnt policies.

In the future, it is possible to extend our framework to model the correlated randomness in achieving different goals with goal-conditioned RL, and in visiting different states with successor representation. We will leverage such informative modeling of the environment in terms of goals and successor states to develop novel RL algorithms in our future work.

## Acknowledgments and Disclosure of Funding

We thank the editor and each reviewer for their constructive comments and suggestions. We thank Tengyu Ma for the valuable suggestion on the MD3QN algorithm.

Funding in direct support of this work: Microsoft.

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
