# A  Appendix

We use Law$(X)$ to denote the distribution of random variable $X$. When $\nu$ is a probability distribution for over set $\Omega$ and $A$ is a subset of $\Omega$, we use $\nu(A)$ to denote the probability that the random variable $X$ belongs to $A$, when $X$ is sampled from distribution $\nu$.

## A.1  Proofs

**Lemma 1** *If $\nu = Law(X)$, then $f_{\#}\nu = Law(f(X))$.*

*Proof.* Since $\nu = \text{Law}(X)$, for all Borel sets $A$, $P(X \in A) = \nu(A)$.

By definition, $f_{\#}\nu(A) = \nu(f^{-1}(A))$ for all Borel sets $A$, and it follows that

$$f_{\#}\nu(A) = P(X \in f^{-1}(A)) = P(f(X) \in A)$$

for all Borel sets $A$.

We thus proved $f_{\#}\nu = \text{Law}(f(X))$. $\qquad\square$

**Lemma 2** *We have*

$$\mathcal{T}^{\pi}\boldsymbol{\mu}(s_t, a_t) = Law(f_{\gamma,\boldsymbol{r}_t}(\boldsymbol{Z})), \text{ i.e. } \mathcal{T}^{\pi}\boldsymbol{\mu}(s_t, a_t) = Law(\gamma \boldsymbol{Z} + \boldsymbol{r}_t)$$

*where $s_{t+1} \sim P(s_{t+1}|s_t, a_t), a_{t+1} \sim \pi(a_{t+1}|s_{t+1}), \boldsymbol{r}_t \sim \boldsymbol{R}(\boldsymbol{r}_t|s_t, a_t)$, and $\boldsymbol{Z} \sim \boldsymbol{\mu}(s_{t+1}, a_{t+1})$.*

*Proof.* For all Borel sets $A$, we have

$$P(f_{\gamma,\boldsymbol{r}_t}(\boldsymbol{Z}) \in A)$$
$$= \int_{\mathbb{R}^N} P(f_{\gamma,\boldsymbol{r}_t}(\boldsymbol{Z}) \in A)\boldsymbol{R}(d\boldsymbol{r}_t|s_t, a_t)$$
$$= \int_{\mathcal{S}} \int_{\mathcal{A}} \int_{\mathbb{R}^N} P(f_{\gamma,\boldsymbol{r}_t}(\boldsymbol{Z}(s_{t+1}, a_{t+1})) \in A)\boldsymbol{R}(d\boldsymbol{r}_t|s_t, a_t)P(ds_{t+1}|s_t, a_t)\pi(da_{t+1}|s_{t+1})$$

where Law$(\boldsymbol{Z}(s_{t+1}, a_{t+1})) = \boldsymbol{\mu}(s_{t+1}, a_{t+1})$. By Lemma 1,

$$\text{Law}(f_{\gamma,\boldsymbol{r}_t}(\boldsymbol{Z}(s_{t+1}, a_{t+1}))) = f_{\gamma,\boldsymbol{r}_t\#}\boldsymbol{\mu}(s_{t+1}, a_{t+1}),$$

it follows that

$$P(f_{\gamma,\boldsymbol{r}_t}(\boldsymbol{Z}) \in A)$$
$$= \int_{\mathcal{S}} \int_{\mathcal{A}} \int_{\mathbb{R}^N} f_{\gamma,\boldsymbol{r}_t\#}\boldsymbol{\mu}(s_{t+1}, a_{t+1})(A)\boldsymbol{R}(d\boldsymbol{r}_t|s_t, a_t)P(ds_{t+1}|s_t, a_t)\pi(da_{t+1}|s_{t+1})$$

for all Borel sets $A$. We then have

$$\text{Law}(f_{\gamma,\boldsymbol{r}_t}(\boldsymbol{Z})) = \int_{\mathcal{S}} \int_{\mathcal{A}} \int_{\mathbb{R}^N} f_{\gamma,\boldsymbol{r}_t\#}\boldsymbol{\mu}(s_{t+1}, a_{t+1})\boldsymbol{R}(d\boldsymbol{r}_t|s_t, a_t)P(ds_{t+1}|s_t, a_t)\pi(da_{t+1}|s_{t+1})$$

and Law$(f_{\gamma,\boldsymbol{r}_t}(\boldsymbol{Z})) = \mathcal{T}^{\pi}\boldsymbol{\mu}(s_t, a_t)$. $\qquad\square$

**Lemma 3** *If $\boldsymbol{r} \in \mathbb{R}^N$, we have*

$$W_p(Law(f_{\gamma,\boldsymbol{r}}(\boldsymbol{Z}_1)), Law(f_{\gamma,\boldsymbol{r}}(\boldsymbol{Z}_2))) \leq \gamma W_p(Law(\boldsymbol{Z}_1), Law(\boldsymbol{Z}_2))$$

*Proof.*

Denote $\nu_1 = \text{Law}(\boldsymbol{Z}_1), \nu_2 = \text{Law}(\boldsymbol{Z}_2)$. Then we have

$$\text{Law}(f_{\gamma,\boldsymbol{r}}(\boldsymbol{Z}_1)) = f_{\gamma,\boldsymbol{r}\#}\nu_1, \text{Law}(f_{\gamma,\boldsymbol{r}}(\boldsymbol{Z}_2)) = f_{\gamma,\boldsymbol{r}\#}\nu_2$$

and Lemma 3 is equivalent to

$$W_p(f_{\gamma,\boldsymbol{r}\#}\nu_1, f_{\gamma,\boldsymbol{r}\#}\nu_2) \leq \gamma W_p(\nu_1, \nu_2). \tag{22}$$

For arbitrary $\epsilon > 0$, suppose $\nu \in \Gamma(\nu_1, \nu_2)$ satisfies that

$$(\int_{\mathbb{R}^N \times \mathbb{R}^N} d(x,y)^p \nu(d^N x, d^N y))^{1/p} < W_p(\nu_1, \nu_2) + \epsilon$$

We consider the distribution $f_{\gamma,\boldsymbol{r}\#}\nu$, which holds that

$$f_{\gamma,\boldsymbol{r}\#}\nu(x,y) = \nu(f_{\gamma,\boldsymbol{r}}^{-1}(x), f_{\gamma,\boldsymbol{r}}^{-1}(y))$$

The marginal distribution on the first $N$ random variables for $f_{\gamma,\boldsymbol{r}\#}\nu$ is

$$\begin{aligned}
f_{\gamma,\boldsymbol{r}\#}\nu(x, \mathbb{R}^N) &= \nu(f_{\gamma,\boldsymbol{r}}^{-1}(x), f_{\gamma,\boldsymbol{r}}^{-1}(\mathbb{R}^N)) \\
&= \nu(f_{\gamma,\boldsymbol{r}}^{-1}(x), \mathbb{R}^N) \\
&= \nu_1(f_{\gamma,\boldsymbol{r}}^{-1}(x)) = f_{\gamma,\boldsymbol{r}\#}\nu_1.
\end{aligned}$$

Similarly, the marginal distribution on the next $N$ random variables for $f_{\gamma,\boldsymbol{r}\#}\nu$ is $f_{\gamma,\boldsymbol{r}\#}\nu_2$. We thus have $f_{\gamma,\boldsymbol{r}\#}\nu \in \Gamma(f_{\gamma,\boldsymbol{r}\#}\nu_1, f_{\gamma,\boldsymbol{r}\#}\nu_2)$.

It follows that

$$\begin{aligned}
W_p(f_{\gamma,\boldsymbol{r}\#}\nu_1, f_{\gamma,\boldsymbol{r}\#}\nu_2) &\leq \int_{\mathbb{R}^N \times \mathbb{R}^N} d(x,y)^p f_{\gamma,\boldsymbol{r}\#}\nu(d^N x, d^N y))^{1/p} \\
&= (\int_{\mathbb{R}^N \times \mathbb{R}^N} d(x,y)^p \nu(d^N(f_{\gamma,\boldsymbol{r}}^{-1}(x)), d^N(f_{\gamma,\boldsymbol{r}}^{-1}(y))))^{1/p} \\
&= (\int_{\mathbb{R}^N \times \mathbb{R}^N} d(f_{\gamma,\boldsymbol{r}}(x), f_{\gamma,\boldsymbol{r}}(y))^p \nu(d^N x, d^N y))^{1/p} \\
&= \gamma(\int_{\mathbb{R}^N \times \mathbb{R}^N} d(x,y)^p \nu(d^N x, d^N y))^{1/p} \\
&= \gamma W_p(\nu_1, \nu_2) + \gamma\epsilon.
\end{aligned}$$

The above equation holds for arbitrary $\epsilon > 0$, thus $W_p(f_{\gamma,\boldsymbol{r}\#}\nu_1, f_{\gamma,\boldsymbol{r}\#}\nu_2) \leq \gamma W_p(\nu_1, \nu_2)$. We thus proved equation (22) and Lemma 3 is proved.

$\square$

**Lemma 4** *Suppose $\boldsymbol{Z}_1(\cdot|A)$ and $\boldsymbol{Z}_2(\cdot|A)$ are two conditional distribution with range $\mathbb{R}^N$, and for all values of $a \in \Omega$,*

$$W_p(Law(\boldsymbol{Z}_1(a)), Law(\boldsymbol{Z}_2(a))) \leq c \tag{23}$$

*where $\boldsymbol{Z}_1(a) \sim \boldsymbol{Z}_1(\cdot|A = a)$, and $\boldsymbol{Z}_2(a) \sim \boldsymbol{Z}_2(\cdot|A = a)$.*

*Then for any distribution $p(A)$ and $A \sim p(A)$, we have*

$$W_p(Law(\boldsymbol{Z}_1), Law(\boldsymbol{Z}_2)) \leq c \tag{24}$$

*where $\boldsymbol{Z}_1 \sim \boldsymbol{Z}_1(\cdot|A)$, $\boldsymbol{Z}_2 \sim \boldsymbol{Z}_2(\cdot|A)$.*

*Proof.* By equation (23), for arbitrary $\epsilon > 0$, there exists $\nu_a$ such that:

For all $a, \nu_a \in \Gamma(Law(\boldsymbol{Z}_1(a)), Law(\boldsymbol{Z}_2(a)))$, and $(\int_{\mathbb{R}^N \times \mathbb{R}^N} d(x,y)^p \nu_a(d^N x, d^N y))^{1/p} \leq c + \epsilon$

We also have

$$Law(\boldsymbol{Z}_1)(\cdot) = \int_\Omega Law(\boldsymbol{Z}_1(a))(\cdot)p(da)$$

We construct distribution $\nu$ such that

$$\nu(x,y) = \int_\Omega \nu_a(x,y)p(da)$$

The marginal distribution on the first $N$ random variables for $\nu$ is

$$\nu(x, \mathbb{R}^N) = \int_\Omega \nu_a(x, \mathbb{R}^N) p(da)$$

$$= \int_\Omega \mathrm{Law}(\boldsymbol{Z}_1)(x) p(da)$$

$$= \mathrm{Law}(\boldsymbol{Z}_1)(x)$$

Similarly, the marginal distribution on the next $N$ random variables for $\nu$ is $\mathrm{Law}(\boldsymbol{Z}_2)(x)$. We thus have $\nu \in \Gamma(\mathrm{Law}(\boldsymbol{Z}_1), \mathrm{Law}(\boldsymbol{Z}_2))$.

It follows that

$$W_p(\mathrm{Law}(\boldsymbol{Z}_1), \mathrm{Law}(\boldsymbol{Z}_2)) \leq (\int_{\mathbb{R}^N \times \mathbb{R}^N} d(x,y)^p \nu(d^N x, d^N y))^{1/p}$$

$$= (\int_{\mathbb{R}^N \times \mathbb{R}^N} d(x,y)^p \int_\Omega \nu_a(d^N x, d^N y) p(da))^{1/p}$$

$$= (\int_\Omega \int_{\mathbb{R}^N \times \mathbb{R}^N} d(x,y)^p \nu_a(d^N x, d^N y) p(da))^{1/p}$$

$$\leq (\int_\Omega (c+\epsilon)^p p(da))^{1/p} = c + \epsilon.$$

The above equation holds for arbitrary $\epsilon > 0$, thus $W_p(\mathrm{Law}(\boldsymbol{Z}_1), \mathrm{Law}(\boldsymbol{Z}_2)) \leq c$. □

**Theorem 1** *For two joint distributions $\boldsymbol{\mu}_1$ and $\boldsymbol{\mu}_2$, we have*
$$\bar{d}_p(\mathcal{T}^\pi \boldsymbol{\mu}_1, \mathcal{T}^\pi \boldsymbol{\mu}_2) \leq \gamma \bar{d}_p(\boldsymbol{\mu}_1, \boldsymbol{\mu}_2). \tag{25}$$

*Proof.* We have $\bar{d}_p(\mathcal{T}^\pi \boldsymbol{\mu}_1, \mathcal{T}^\pi \boldsymbol{\mu}_2) = \sup_{s_t, a_t} W_p(\mathcal{T}^\pi \boldsymbol{\mu}_1(s_t, a_t), \mathcal{T}^\pi \boldsymbol{\mu}_2(s_t, a_t))$.

For each $(s_t, a_t)$, we let $s_{t+1} \sim P(s_{t+1}|s_t, a_t), a_{t+1} \sim \pi(a_{t+1}|s_{t+1}), \boldsymbol{r}_t \sim \boldsymbol{R}(\boldsymbol{r}_t|s_t, a_t), \boldsymbol{Z}_1 \sim \boldsymbol{\mu}_1(s_{t+1}, a_{t+1})$ and $\boldsymbol{Z}_2 \sim \boldsymbol{\mu}_2(s_{t+1}, a_{t+1})$.

By Lemma 2, we have
$$W_p(\mathcal{T}^\pi \boldsymbol{\mu}_1(s_t, a_t), \mathcal{T}^\pi \boldsymbol{\mu}_2(s_t, a_t)) = W_p(\mathrm{Law}(f_{\gamma, \boldsymbol{r}_t}(\boldsymbol{Z}_1)), \mathrm{Law}(f_{\gamma, \boldsymbol{r}_t}(\boldsymbol{Z}_2)))$$

By Lemma 3,
$$W_p(\mathrm{Law}(f_{\gamma, \boldsymbol{r}}(\boldsymbol{Z}_1)), \mathrm{Law}(f_{\gamma, \boldsymbol{r}}(\boldsymbol{Z}_2))) \leq \gamma W_p(\mathrm{Law}(\boldsymbol{Z}_1), \mathrm{Law}(\boldsymbol{Z}_2))$$

for each constant $\boldsymbol{r} \in \mathbb{R}^N$. By Lemma 4, when $\boldsymbol{r}_t \sim \boldsymbol{R}(\boldsymbol{r}_t|s_t, a_t)$, we have
$$W_p(\mathrm{Law}(f_{\gamma, \boldsymbol{r}_t}(\boldsymbol{Z}_1)), \mathrm{Law}(f_{\gamma, \boldsymbol{r}_t}(\boldsymbol{Z}_2))) \leq \gamma W_p(\mathrm{Law}(\boldsymbol{Z}_1), \mathrm{Law}(\boldsymbol{Z}_2)).$$

Since for each $(s_{t+1}, a_{t+1})$,
$$W_p(\boldsymbol{\mu}_1(s_{t+1}, a_{t+1}), \boldsymbol{\mu}_2(s_{t+1}, a_{t+1})) \leq \sup_{s,a} W_p(\boldsymbol{\mu}_1(s, a), \boldsymbol{\mu}_2(s, a)),$$

by Lemma 4, when $s_{t+1} \sim P(s_{t+1}|s_t, a_t), a_{t+1} \sim \pi(a_{t+1}|s_{t+1})$, we have
$$W_p(\mathrm{Law}(\boldsymbol{Z}_1), \mathrm{Law}(\boldsymbol{Z}_2)) \leq \sup_{s,a} W_p(\boldsymbol{\mu}_1(s, a), \boldsymbol{\mu}_2(s, a))$$

where $\boldsymbol{Z}_1 \sim \boldsymbol{\mu}_1(s_{t+1}, a_{t+1})$ and $\boldsymbol{Z}_2 \sim \boldsymbol{\mu}_2(s_{t+1}, a_{t+1})$.

Combining all the above derivations, we have
$$W_p(\mathcal{T}^\pi \boldsymbol{\mu}_1(s_t, a_t), \mathcal{T}^\pi \boldsymbol{\mu}_2(s_t, a_t)) = W_p(\mathrm{Law}(f_{\gamma, \boldsymbol{r}_t}(Z_1)), \mathrm{Law}(f_{\gamma, \boldsymbol{r}_t}(Z_2)))$$
$$\leq \gamma W_p(\mathrm{Law}(\boldsymbol{Z}_1), \mathrm{Law}(\boldsymbol{Z}_2))$$
$$\leq \gamma \sup_{s,a} W_p(\boldsymbol{\mu}_1(s, a), \boldsymbol{\mu}_2(s, a)),$$

and
$$\bar{d}_p(\mathcal{T}^\pi \boldsymbol{\mu}_1, \mathcal{T}^\pi \boldsymbol{\mu}_2) = \sup_{s_t, a_t} W_p(\mathcal{T}^\pi \boldsymbol{\mu}_1(s_t, a_t), \mathcal{T}^\pi \boldsymbol{\mu}_2(s_t, a_t))$$
$$\leq \gamma \sup_{s,a} W_p(\boldsymbol{\mu}_1(s, a), \boldsymbol{\mu}_2(s, a)) = \gamma \bar{d}_p(\boldsymbol{\mu}_1, \boldsymbol{\mu}_2).$$

□

**Lemma 5** $\boldsymbol{\mu}^\pi$ *is the fixed-point of the operator* $\mathcal{T}^\pi$.

*Proof.* By definition,

$$\boldsymbol{\mu}^\pi(s, a) = \text{Law}(\sum_{t=0}^{\infty} \gamma^t \boldsymbol{r}_t),$$

where $s_0 = s, a_0 = a, \boldsymbol{r}_t \sim R(\cdot|s_t, a_t), s_{t+1} \sim P(\cdot|s_t, a_t), a_{t+1} \sim \pi(\cdot|s_{t+1})$.

which is equivalent to

$$\boldsymbol{\mu}^\pi(s_t, a_t) = \text{Law}(\sum_{\tau=t}^{\infty} \gamma^{\tau-t} \boldsymbol{r}_\tau), \tag{26}$$

where $\boldsymbol{r}_\tau \sim R(\cdot|s_\tau, a_\tau), s_{\tau+1} \sim P(\cdot|s_\tau, a_\tau), a_{\tau+1} \sim \pi(\cdot|s_{\tau+1})$ for all $\tau \geq t$.

By Lemma 2, we have

$$\mathcal{T}^\pi \boldsymbol{\mu}^\pi(s_t, a_t) = \text{Law}(\gamma \boldsymbol{Z} + \boldsymbol{r}_t)$$

where $s_{t+1} \sim P(s_{t+1}|s_t, a_t), a_{t+1} \sim \pi(a_{t+1}|s_{t+1}), \boldsymbol{r}_t \sim \boldsymbol{R}(\boldsymbol{r}_t|s_t, a_t)$, and $\boldsymbol{Z} \sim \boldsymbol{\mu}^\pi(s_{t+1}, a_{t+1})$.

Substituting $\boldsymbol{\mu}^\pi(s_{t+1}, a_{t+1})$ by equation (26), we have

$$\mathcal{T}^\pi \boldsymbol{\mu}^\pi(s_t, a_t) = \text{Law}(\gamma \sum_{\tau=t+1}^{\infty} \gamma^{\tau-(t+1)} \boldsymbol{r}_\tau + \boldsymbol{r}_t) = \text{Law}(\sum_{\tau=t}^{\infty} \gamma^{\tau-t} \boldsymbol{r}_\tau)$$

where $\boldsymbol{r}_\tau \sim R(\cdot|s_\tau, a_\tau), s_{\tau+1} \sim P(\cdot|s_\tau, a_\tau), a_{\tau+1} \sim \pi(\cdot|s_{\tau+1})$ for all $\tau \geq t$.

We then proved that $\mathcal{T}^\pi \boldsymbol{\mu}^\pi = \boldsymbol{\mu}^\pi$. $\qquad\square$

**Corollary 1** *If* $\boldsymbol{\mu}_{i+1} = \mathcal{T}^\pi \boldsymbol{\mu}_i$, *then as* $i \to \infty$, $\boldsymbol{\mu}_i \to \boldsymbol{\mu}^\pi$.

*Proof.* Since $\mathcal{T}^\pi$ is a contraction in $\bar{d}_p$, by Banach's fixed point theorem, there is one and only one fixed point of $\mathcal{T}^\pi$, and as $i \to \infty$, $\boldsymbol{\mu}_i$ converges to this fixed point. By Lemma 5, $\boldsymbol{\mu}^\pi$ is the only fixed point of $\mathcal{T}^\pi$, and as $i \to \infty$, $\boldsymbol{\mu}_i \to \boldsymbol{\mu}^\pi$. $\qquad\square$

**Lemma 6** *We have*

$$\mathcal{T}\boldsymbol{\mu}(s_t, a_t) = Law(f_{\gamma, \boldsymbol{r}_t}(\boldsymbol{Z})), \tag{27}$$

*where* $\boldsymbol{r}_t \sim \boldsymbol{R}(\boldsymbol{r}_t|s_t, a_t)$, $s_{t+1} \sim P(s_{t+1}|s_t, a_t)$, $a' = \arg\max_a \mathbb{E}_{\boldsymbol{Z} \sim \boldsymbol{\mu}(s_{t+1}, a)} \sum_{n=1}^{N} Z_n$, *and* $\boldsymbol{Z} \sim \boldsymbol{\mu}(s_{t+1}, a')$.

*Proof.* For all Borel sets $A$, we have

$$P(f_{\gamma, \boldsymbol{r}_t}(\boldsymbol{Z}) \in A)$$
$$= \int_{\mathbb{R}^N} P(f_{\gamma, \boldsymbol{r}_t}(\boldsymbol{Z}) \in A) \boldsymbol{R}(d\boldsymbol{r}_t|s_t, a_t)$$
$$= \int_S \int_{\mathbb{R}^N} P(f_{\gamma, \boldsymbol{r}_t}(\boldsymbol{Z}(s_{t+1}, a')) \in A) \boldsymbol{R}(d\boldsymbol{r}_t|s_t, a_t) P(ds_{t+1}|s_t, a_t)$$

where $a' = \arg\max_a \mathbb{E}_{\boldsymbol{Z} \sim \boldsymbol{\mu}(s_{t+1}, a)} \sum_{n=1}^{N} Z_n$.

By Lemma 1,

$$\text{Law}(f_{\gamma, \boldsymbol{r}_t}(\boldsymbol{Z}(s_{t+1}, a'))) = f_{\gamma, \boldsymbol{r}_t \#} \boldsymbol{\mu}(s_{t+1}, a'),$$

it follows that

$$P(f_{\gamma, \boldsymbol{r}_t}(\boldsymbol{Z}) \in A) = \int_S \int_{\mathbb{R}^N} f_{\gamma, \boldsymbol{r}_t \#} \boldsymbol{\mu}(s_{t+1}, a')(A) \boldsymbol{R}(d\boldsymbol{r}_t|s_t, a_t) P(ds_{t+1}|s_t, a_t)$$

for all Borel sets $A$, and

$$\text{Law}(f_{\gamma, \boldsymbol{r}_t}(\boldsymbol{Z})) = \int_S \int_{\mathbb{R}^N} f_{\gamma, \boldsymbol{r}_t \#} \boldsymbol{\mu}(s_{t+1}, a') \boldsymbol{R}(d\boldsymbol{r}_t|s_t, a_t) P(ds_{t+1}|s_t, a_t) = \mathcal{T}\boldsymbol{\mu}(s_t, a_t)$$

$$\square$$

**Theorem 2** *For two joint distributions $\boldsymbol{\mu}_1$ and $\boldsymbol{\mu}_2$, we have*

$$||(\mathbb{E}_{\sum})(\mathcal{T}\boldsymbol{\mu}_1) - (\mathbb{E}_{\sum})(\mathcal{T}\boldsymbol{\mu}_2)||_\infty \le \gamma||(\mathbb{E}_{\sum})\boldsymbol{\mu}_1 - (\mathbb{E}_{\sum})\boldsymbol{\mu}_2||, \tag{28}$$

*Proof.* We first prove that

$$(\mathbb{E}_{\sum})(\mathcal{T}\boldsymbol{\mu}) = \mathcal{T}_E(\mathbb{E}_{\sum})\boldsymbol{\mu} \tag{29}$$

where $\mathcal{T}_E$ is the Bellman optimality operator over scalar values.

We have

$$(\mathbb{E}_{\sum})(\mathcal{T}\boldsymbol{\mu})(s_t, a_t) = \mathbb{E}_{\boldsymbol{Z}\sim\boldsymbol{\mu}(s_{t+1},a')} \sum_{n=1}^{N}(f_{\gamma,\boldsymbol{r}_t}(\boldsymbol{Z}))_n$$

where $\boldsymbol{r}_t \sim \boldsymbol{R}(\boldsymbol{r}_t|s_t, a_t)$, $s_{t+1} \sim P(s_{t+1}|s_t, a_t)$, and $a' = \arg\max_a \mathbb{E}_{\boldsymbol{Z}\sim\boldsymbol{\mu}(s_{t+1},a)} \sum_{n=1}^{N} Z_n$.

It follows that

$$\begin{aligned}
(\mathbb{E}_{\sum})(\mathcal{T}\boldsymbol{\mu})(s_t, a_t) &= \sum_{n=1}^{N}(r_t)_n + \gamma \mathbb{E}_{\boldsymbol{Z}\sim\boldsymbol{\mu}(s_{t+1},a')} \sum_{n=1}^{N} Z_n \\
&= r_t + \gamma \max_{a'} \mathbb{E}_{\boldsymbol{Z}\sim\boldsymbol{\mu}(s_{t+1},a')} \sum_{n=1}^{N} Z_n \\
&= r_t + \gamma \max_{a'}(\mathbb{E}_{\sum})\boldsymbol{\mu}(s_{t+1}, a') \\
&= \mathcal{T}_E(\mathbb{E}_{\sum})\boldsymbol{\mu}(s_t, a_t)
\end{aligned}$$

Then we have

$$\begin{aligned}
||(\mathbb{E}_{\sum})(\mathcal{T}\boldsymbol{\mu}_1) - (\mathbb{E}_{\sum})(\mathcal{T}\boldsymbol{\mu}_2)||_\infty &= ||\mathcal{T}_E(\mathbb{E}_{\sum})\boldsymbol{\mu}_1 - \mathcal{T}_E(\mathbb{E}_{\sum})\boldsymbol{\mu}_2||_\infty \\
&\le \gamma||(\mathbb{E}_{\sum})\boldsymbol{\mu}_1 - (\mathbb{E}_{\sum})\boldsymbol{\mu}_2||_\infty
\end{aligned}$$

$\square$

**Corollary 2** *If $\boldsymbol{\mu}_{i+1} = \mathcal{T}\boldsymbol{\mu}_i$, then as $i \to \infty$, $\mathbb{E}_{\boldsymbol{Z}\sim\boldsymbol{\mu}_i(s,a)} \sum_{n=1}^{N} Z_n \to Q^*(s,a)$ for all $(s,a)$.*

*Proof.* Since $\mathcal{T}_E$ is a contraction on $\infty$-norm and $Q^*$ is the fixed point of $\mathcal{T}_E$, by Banach's fixed point theorem, $Q^*$ is the only fixed point of $\mathcal{T}_E$, and $\mathcal{T}_E^i(\mathbb{E}_{\sum})\boldsymbol{\mu}_0 \to Q^*$ as $i \to \infty$.

By equation (29), $(\mathbb{E}_{\sum})\boldsymbol{\mu}_i = \mathcal{T}_E(\mathbb{E}_{\sum})\boldsymbol{\mu}_{i-1} = \cdots = \mathcal{T}_E^i(\mathbb{E}_{\sum})\boldsymbol{\mu}_0$.

Therefore, as $i \to \infty$, $(\mathbb{E}_{\sum})\boldsymbol{\mu}_i \to Q^*$, and $\mathbb{E}_{\boldsymbol{Z}\sim\boldsymbol{\mu}_i(s,a)} \sum_{n=1}^{N} Z_n \to Q^*(s,a)$ for all $(s,a)$. $\square$

### A.2 Environment Settings and Hyperparameter Settings

**Atari environment settings.** The environment settings for Atari environments are shown in Table 1. We use six games on Atari: AirRaid, Asteroids, Pong, MsPacman, Gopher and UpNDown. The reward functions for these environments are set as follows.

- For AirRaid, the agent gets multi-dimensional reward $[100, 0, 0, 0]$, $[0, 75, 0, 0]$, $[0, 0, 50, 0]$, $[0, 0, 0, 25]$, $[0, 0, 0, 0]$ respectively for the primitive reward 100, 75, 50, 25 and 0;
- For Pong, if the primitive reward for the agent is $-1$, the agent gets a multi-dimensional reward $[-1, 0]$; if the primitive reward for the agent is 1, the agent gets a multi-dimensional reward $[0, 1]$; otherwise, the agent gets a multi-dimensional reward $[0, 0]$.
- For Asteroids, we denote the primitive reward as $r$, and denote the multi-dimensional reward as $[r_1, r_2, r_3]$. If $(r - 20) \mod 50 = 0$, we let $r_1 = 20$, otherwise $r_1 = 0$. If $(r - r_1 - 50) \mod 100 = 0$, we let $r_2 = 50$, otherwise $r_2 = 0$. We let $r_3 = r - r_1 - r_2$.
- For MsPacman, we denote the primitive reward as $r$, and denote the multi-dimensional reward as $[r_1, r_2, r_3, r_4]$. If $(r - 10) \mod 50 = 0$, we let $r_1 = 10$, otherwise $r_1 = 0$. If $(r - r_1 - 50) \mod 100 = 0$, we let $r_2 = 50$, otherwise $r_2 = 0$. If $(r - r_1 - r_2 - 100) \mod 200 = 0$, we let $r_3 = 100$, otherwise $r_3 = 0$. We let $r_4 = r - r_1 - r_2 - r_3$.

- For Gopher, we denote the primitive reward as $r$, and denote the multi-dimensional reward as $[r_1, r_2]$. If $(r - 20) \mod 100 = 0$, we let $r_1 = 20$, otherwise $r_1 = 0$. We let $r_2 = r - r_1$.

- For UpNDown, we denote the primitive reward as $r$, and denote the multi-dimensional reward as $[r_1, r_2, r_3]$. If $(r - 10) \mod 100 = 0$, we let $r_1 = 10$, otherwise $r_1 = 0$. If $(r - r_1 - 100) \mod 200 = 0$, we let $r_2 = 100$, otherwise $r_2 = 0$. We let $r_3 = r - r_1 - r_2$.

**Maze environment settings.** The environment settings for maze environments are shown in Table 1. The observations for maze are $84 \times 84 \times 3$ RGB images. The reward functions for three maze environments are set as follows.

- For "maze-exclusive" and "maze-identical" environment, each position with a green square awards the agent for $r \sim U(0.2, 0.6)$ in the first reward source; each position with a red square awards the agent for $r \sim U(0.4, 0.9)$ in the second reward source.

- For "maze-multireward" environment, the orange square awards the agent for $r \sim U(0.2, 0.4)$ in the first reward source; the blue square awards the agent for $r \sim U(0.8, 1.0)$ in the second reward source; the green square awards the agent for $r \sim U(0.3, 0.5)$ in the third reward source; the red square awards the agent for $r \sim U(0.5, 0.7)$ in the fourth reward source.

| Environment settings | Values for Maze | Values for Atari |
|---|---|---|
| Stack size | 1 | 4 |
| Frame skip | 1 | 4 |
| One-frame observation shape | (84, 84, 3) | (84, 84) |
| Agent's observation shape | (84, 84, 3) | (84, 84, 4) |
| $\gamma$ | 0.99 | 0.99 |
| Reward clipping | — | true |
| Terminate on Life Loss | — | true |
| Sticky Actions | false | false |

Table 1: Environment settings for maze and Atari.

**MD3QN settings.** The hyper parameter settings for MD3QN is provided in Table 2. For the maze environment, we set the bandwidths to be $[0.01, 0.02, 0.04, 0.08, 0.16, 0.32, 0.48, 0.64, 0.80, 0.96]$; for Atari environment, we use the same bandwidth settings as MMDQN.

| Hyper-parameters | Values for MD3QN |
|---|---|
| Learning Rate | 5e-5 |
| Optimizer | Adam |
| Squared bandwidth | $[2^{-8}, 2^{-7}, 2^{-6}, \cdots, 2^8]$ |
| Network architecture | Refer to section 3.4 |
| Number of particles ($M$) | 200 |
| $\epsilon$-train | Linear decay from 1 to 0.01 |
| $\epsilon$-evaluation | 0.001 |
| decay period for $\epsilon$ | 1M frames |

Table 2: Hyper-parameter settings for MD3QN

## A.3 Additional Experimental Results

### A.3.1 Joint Distribution Modeling on Maze Environment

Under the maze environment and policy evaluation setting, we visualize the joint distribution throughout the training process for the experiment in section 5.1, and the result is shown in Figure 4.

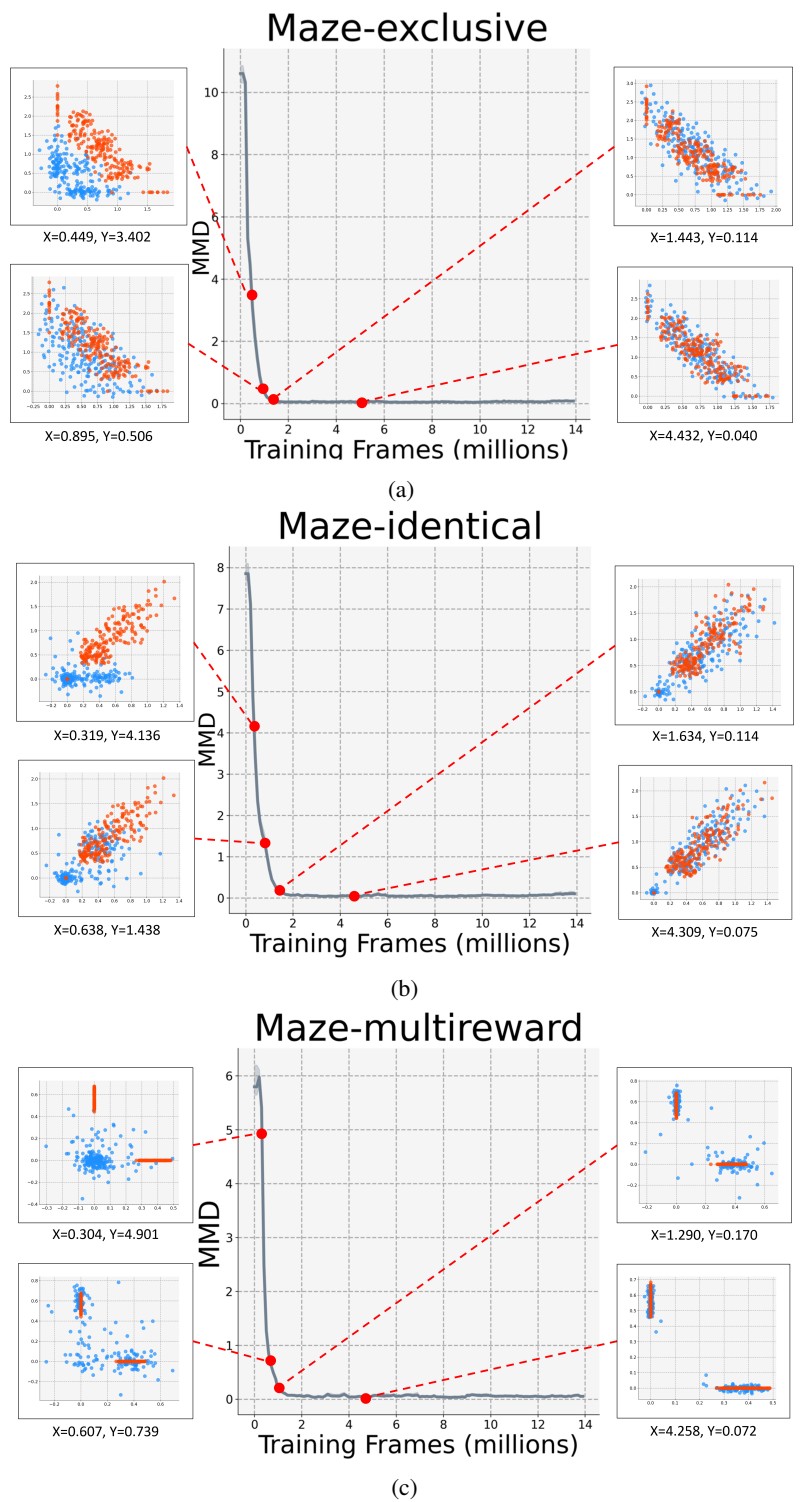

Figure 4: Visualization of the modeled joint distribution throughout the training process under three maze environments. The y-axis represents the MMD metric between the modeled joint distribution (blue dots) and the true distribution $\boldsymbol{\mu}^{\pi}$ (red dots), which we use only for evaluation purpose.

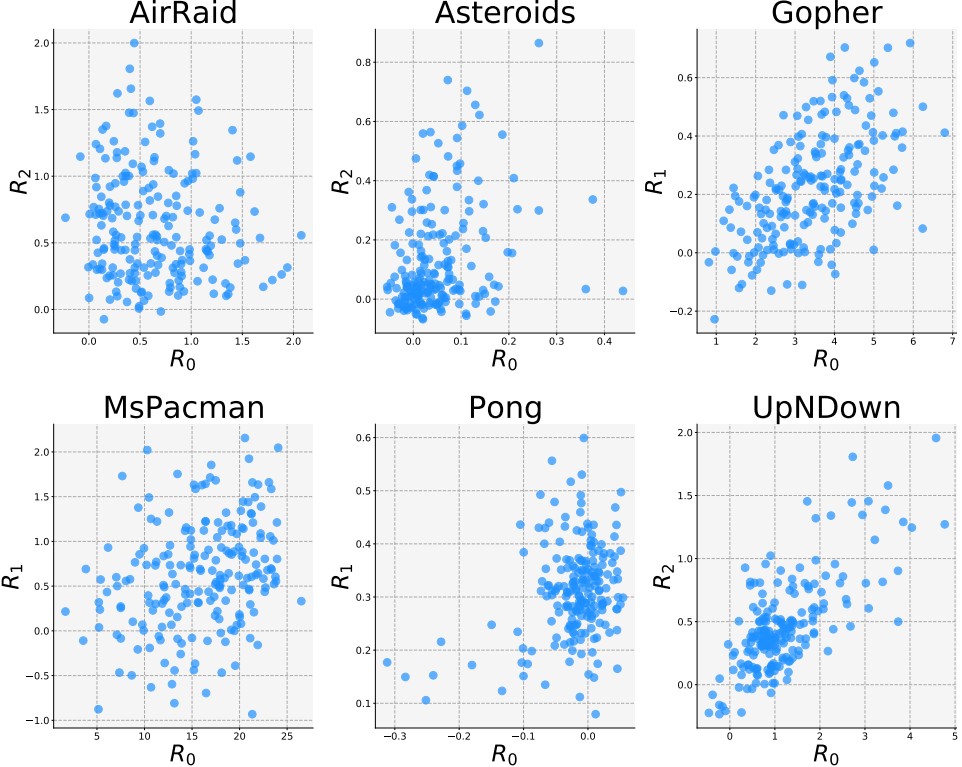

Figure 5: Samples of joint distribution by MD3QN on six Atari games. For each game, we show the joint distribution for two dimensions of reward for visualization purpose.

### A.3.2 Joint Distribution Modeling on Atari Games

For the case studies for the modeled joint distributional by MD3QN on Atari games, we provide the results in Figure 5. Overall, both the reward correlation and the reward scale are reasonably captured by MD3QN. In the aspect of reward correlation, Gopher's two sources of reward are positively correlated, since the agent needs to remove holes on the ground in order to kill the monster; UpNDown's two sources of reward are also positively correlated, since the reward for being alive and the reward for killing enemy cars have positive correlation. In the aspect of reward scale, the samples generated by MD3QN align with the scale of the true return distribution in all dimensions.

### A.3.3 Downstream task for MD3QN: RL with multiple constraints

We add another experiment to show the necessity of modeling the joint distribution, and explain why only modeling marginal distributions may fail in some settings (e.g., in environments with multiple correlated constraints) from theory and experiment results.

There are a number of real-world scenarios where multiple constraints should be met simultaneously. For instance, in autonomous driving, we need to balance the safety (distance from other cars), the speed, the comfort (the acceleration, etc.), and many other factors to make sure the car functions normally. Those statistics can be viewed as sub-rewards in our settings. For instance, we would like to make the agent be aware of the joint distributions of distance from other cars and the speed, and add a constraint on this distribution (e.g., $d > 1$m, $v < 50$km/h).

From the theoretical perspective, only the algorithm which is capable of modeling the joint distribution can find this optimal solution. If the algorithm can only model the marginal distributions, we can correctly compute the probability of simultaneously meeting multiple constraints only if they are independent. However, this is not common in real-world scenarios.

We use the same Maze environment as in our paper and modify the layout as a simplified analogue (see Figure 6(b)). We set multiple constraints on the total return (in our experiment, we have three

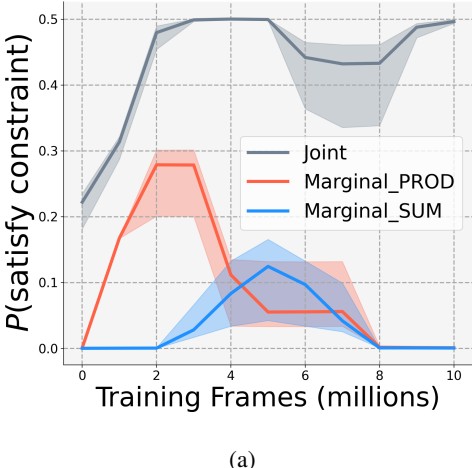 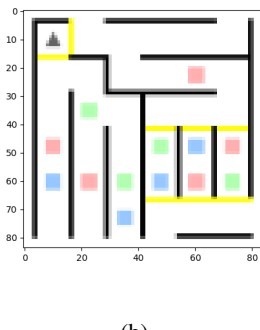

(a)
(b)

Figure 6: (a): maze environment with multiple constraints. (b): probability of the agent to satisfy all constraints during evalution throughout the training process.

constraints: for the initial state, total red return $> 0.6$, total green return $> 0.6$, and total blue return $> 0.6$, where the agent gets a 1.0 reward of the specific color after collecting a specific colored block), and the goal is to find a policy which has the highest probability to satisfy all constraints.

We use the joint distribution by MD3QN to achieve this: specifically, given the modeled joint distribution $\boldsymbol{\mu}(s, a)$, the agent can compute the probability to satisfy all the constraints in the joint distribution and take action by $\arg\max_a \mathbb{P}_{\boldsymbol{Z}\sim\boldsymbol{\mu}(s,a)}(\boldsymbol{Z}$ satisfy all three constraints$)$. We also modify the joint distributional Bellman optimality operator to maximize the probability to satisfy all constraints.

For baseline methods, we extend the MMDQN by multiple heads $\mu_{1:N}$ to reflect $N$ sources of rewards. This method is different from our algorithm in that only the marginal distribution of each dimension is learned. We test two ways to approximately compute the probability to satisfy all constraints by marginal distributions: the "Marginal_SUM" baseline maximizes $\sum_{i=1}^{N} \mathbb{P}_{Z_i\sim\mu_i(s,a)}(Z_i$ satisfy $i$-th constraint$)$, the sum of probabilities for each reward to satisfy constraint, while the "Marginal_PROD" baseline maximizes $\Pi_{i=1}^{N}\mathbb{P}_{Z_i\sim\mu_i(s,a)}(Z_i$ satisfy $i$-th constraint$)$, the product of probabilities for each reward to satisfy constraint.

The results is shown in Figure 6(a). It can be seen that maximizing the probability based on joint distribution can significantly outperform two baseline methods which both use marginal distribution information.

### A.3.4 Effect of Network Architectures on Joint Distribution Modeling

We study the impact of the choice of network architecture to the accuracy of joint return distribution modeling in MD3QN, and illustrate why we use the network architecture in section 3.4.

We test the performance of several network architectures on policy evaluation setting at the beginning before fixing the architecture:

1. The architecture used in the current paper. To be specific, we replace the final layer of DQN to output some deterministic samples of N-dimensional return for each action (see section 3.4).

2. An IQN like architecture to model the joint distribution which multiplies the state features with the cosine embedding of a uniform noise signal, and outputs the multi-dimensional return samples.

3. A network architecture that concatenates a noise to the state feature, and outputs the deterministic samples.

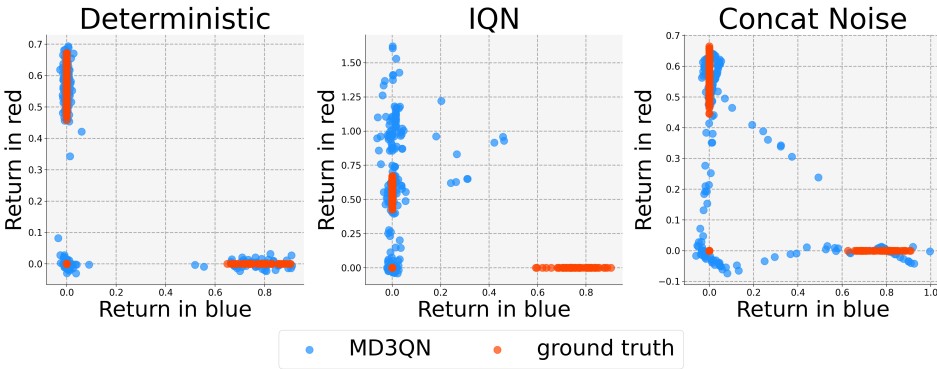

Figure 7: Results for joint return distribution modeling under three network architectures. The experiment settings are the same as in Section 5.1.

We perform the same experiment as in Section 5.1 and Figure 2 in our paper (under maze environments and policy evaluation setting) to test how MD3QN models the joint distribution under these three network architectures. The results is shown in Figure 7.

For the first architecture, the MMD between prediction and ground truth distribution is 0.026, while for the second architecture, MMD is 1.541, and for the third architecture, MMD is 0.204. It can be concluded that the first one is the best in terms of modeling the joint distribution, and we adopt this architecture in our paper.

### A.3.5    Effect of Bandwidth Selection on Joint Distribution Modeling

We study the impact of the choice of bandwidth selection to the accuracy of joint return distribution modeling in MD3QN, and illustrate how we select the bandwidth for our experiments.

We tested 3 sets of squared kernel bandwidths on policy evaluation setting:

- Squared bandwidths $W_1 = [2^{-8}, 2^{-7}, 2^{-6}, \cdots, 2^8]$
- Squared bandwidths $W_2 = [0.01, 0.02, 0.04, 0.08, 0.16, 0.32, 0.48, 0.64, 0.80, 0.96, 2.0, 5.0, 10.0, 20.0, 50.0, 100.0]$
- Squared bandwidths $W_3 = [1, 2, 3, 4, 5, 6, 7, 8, 9, 10]$

For any set of squared bandwidths $W = [w_1, w_2, \cdots, w_M]$, the kernel function is given by $k(x, x') = \sum_{i=1}^{M} \exp(-\frac{1}{w_i} \|x - x'\|_2^2)$, which is used to define the MMD metric.

We perform the same experiment as in Section 5.1 and Figure 2 in our paper (under maze environments and policy evaluation setting) to find how accurately MD3QN models the joint distribution under different bandwidth settings. The result can is shown in Figure 8. For squared bandwidths $W_1$, the MMD metric between prediction and ground truth distribution is 0.026, for $W_2$, MMD = 0.036, and for $W_3$, MMD = 1.348. We find that $W_1$ and $W_2$ can capture the joint distribution better than $W_3$, and $W_1$ is slightly better than $W_2$. Finally, we use squared bandwidth $W_1$ in our experiments.

In general, we conclude that the kernel bandwidths should cover a wide range, which helps MD3QN to precisely capture the true joint distribution.

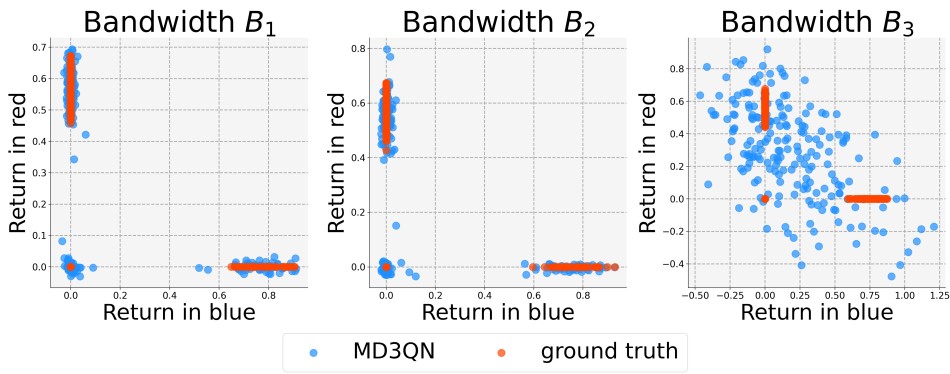

Figure 8: Results for joint return distribution modeling under three sets of bandwidths. The experiment settings are the same as in Section 5.1.