# OpenReview forum: "Distributional Reinforcement Learning for Multi-Dimensional Reward Functions"
_NeurIPS.cc/2021/Conference — NeurIPS 2021 Poster_

### Official Review · Reviewer_TpSB · 2021-07-15

**Rating:** 5
**Confidence:** 5

**Summary:**

This paper proposes a method for distributional reinforcement in environments with multiple channels of reward. The joint distributional Bellman operator is introduced and its properties analysed, and an implementation based on MMD distances is described, with experiments carried on a small-scale gridworld and several Atari games.

**Limitations And Societal Impact:**

See main review for comments on limitations. I did not see any discussion around societal impact, although this work is concerned with a core reinforcement learning algorithm, rather than applications.

**Main Review:**

__Post-discussion__

I thank the authors for their detailed rebuttal. I like the fundamental idea of the paper, although I think that the updates required to the paper are substantial enough that further review may be beneficial, particularly around the unbiasedness claims for MMD estimation, "common misunderstanding" remarks, and experimental details. Based on this, I would not recommend acceptance at this time, but I think that with further updates along these lines, this will be a strong paper.

__Main points__

There is an interesting central contribution in this paper, that I think should be of interest broadly to the RL community. Both distributional reinforcement learning and multi-reward channel RL are existing ideas, but their combination is an interesting idea, and the perspective the authors take in this paper is novel as far as I am aware. The exposition is generally clear, and the experimental results show interesting results from the newly-proposed approach. I would therefore like to see this paper accepted.

However, there are several parts of the paper that I think require clarification before the paper is ready for acceptance:

1. I found the discussion in the paragraphs at Lines 107 & 160 very unclear, for several reasons:
      * First, it is known that metrics for analysis and for projection play different roles in distributional reinforcement learning; see for example the analysis of Dabney et al. (2018b), in which a W_1 projection is used, while the contraction of the projected Bellman operator is performed under W_infty.
      * Second, the paragraph at Line 160 is unclear when mentioning "convergence" as to what it is referring to.  Repeated application of the distributional Bellman operator (or joint distributional Bellman operator) can be shown to be convergent using a variety of metrics, but this is distinct from understanding convergence of approximate algorithms. The example of C51 mentioned in this paragraph is useful to consider: global minimization of the expected KL loss leads to the same result as the application of the projected Bellman operator, this does not imply that using the KL loss leads to a convergent algorithm. Analysis of the projected operator, for example, would be required to show this. On this basis, I don't think the sentence "We illustrate that, once the convergence of the Bellman operator is proved under a certain metric, we can use other metrics than the metric for proving convergence, as training loss to approximate the Bellman operator." is true.
2. I think currently there is not enough discussion of the fact that multiple independent samples from T mu (s, a; \theta_i) are required to unbiasedly estimate gradients of the MMD^2 loss. This is an important fact, since multiple independent samples from a single given state-action pair are typically not available in online/replay-based reinforcement learning (e.g. Atari), which makes the sentence "the temporal difference loss MMD^2 can be optimized without bias by transition samples" potentially misleading.
3. Experiments: Overall these experiments nicely demonstrate the methods introduced by the paper, but there are some missing details that should be added. I couldn't find values for some hyperparameters such as the number of particles used, and in other cases I couldn't find details of how hyperparameter sweeps (e.g. on kernel bandwidth, learning rates, etc.) were carried out, or on how the Atari environments were selected, or their multichannel reward functions were designed (i.e. were other alternative rewards/games tried, did these work as well, etc.)

__Minor points__

Proposition 1: The metric of convergence should be specified; presumably it is \overline{d}_p.

There is some previous work on multi-channel distributional reinforcement learning that it would be good to comment on in the related work section, such as "Distributional Reward Decomposition for Reinforcement Learning" (NeurIPS, 2019).

The paper would naturally be stronger with an evaluation carried out on the full set of Atari games. However, I don't think this should be necessary for acceptance. The experiments that are provided in the paper serve as a useful proof-of-concept that the approach can learn meaningful joint distributions over rewards, and that this approach can have benefits in large-scale RL agents.

__Notation/typos__

E_\Sigma vs \mathbb{E}_\Sigma
"Guassian" -> "Gaussian"

**Time Spent Reviewing:**

4

---

> ### Author Response · Authors · 2021-08-10
> **Response to Reviewer TpSB**
>
> Thank you for the thoughtful and constructive suggestions! We have taken all the comments into consideration and summarized the responses as follows:
>
> 1) *I found the discussion in the paragraphs at Lines 107 & 160 very unclear, for several reasons:*
>
>     a) *First, it is known that metrics for analysis and for projection play different roles in distributional reinforcement learning;*
>
>     b) *Second, the paragraph at Line 160 is unclear when mentioning "convergence" as to what it is referring to.*
>
> **Response:**
> Thanks for your advice! We are sorry that this point was not clear in our original manuscript, and we apologize for all the confusion.
>
> For the first comment, we believe that maybe it's not that appropriate to claim “common misunderstanding”. However, there do exist some expressions in the existing work that are kind of misleading.
>
> In [1], they raise “an open question about the sufficient condition for contraction”, suggesting that they think that they need to prove contraction under MMD metric since they use MMD as their loss. However, the convergence result is already built with the contraction result under the Wasserstein metric in [2]. Based on the aforementioned works, we believe that there does exist some misleading information in existing work.
>
> For the second comment, we agree that the expression in the original manuscript is not rigorous enough. We would like to restate our claim: (1) we can use a metric A to prove that the (projected) distributional Bellman operator is a contraction, and (2) we can use another distribution measure B as training loss, as long as global minimization of this loss leads to the same result as the application of the (projected) Bellman operator.
>
> To be specific, for the distributional Bellman operator, the metric A used for proof and measure B used as loss are not necessarily the same. In our case, we use MMD loss as B, and the above claim holds when the number of samples goes to infinity. For the projected distributional Bellman operator, there exists another metric used for projection, and this is not necessarily the same as B, too.
>
> Since (projected) Bellman operator with metric A is a contraction, and global minimization of loss B has the same result (projected) Bellman operator, it is natural to conclude that it is okay to use B as training loss with the theoretical guarantee proved by A not broken.
>
> [1] Distributional Reinforcement Learning via Moment Matching.
>
> [2] A Distributional Perspective on Reinforcement Learning.
>
> -------------------------------------------------------------------------------------
>
> 2) *I think currently there is not enough discussion of the fact that multiple independent samples from T mu (s, a; \theta_i) are required to unbiasedly estimate gradients of the MMD^2 loss.*
>
> **Response:**
> In lines 193-194 of our paper, we write a potentially inaccurate expression which may cause misunderstanding, which says that “the second term of equation (18) can be unbiasedly estimated by drawing independent samples from joint distribution µ(s, a; θi+1) and from Tµ(s, a; θi)”. By “independent samples” here, we mean that each sample from µ(s, a; θi+1) is independent of every sample from Tµ(s, a; θi), not that samples from Tµ(s, a; θi) should be independent of each other. The second term in equation (18) also reflects this, where we only require two random variables $x \sim p$ and $y \sim q$ to be independent, without requiring different samples from $q$ to be independent. Therefore, our algorithm can use only one transition sample to generate multiple dependent samples of Tµ(s, a; θi), and estimate the gradient of MMD^2 without bias as in lines 5-6 of Algorithm 1. Despite that, we agree that lines 193-194 in our paper are not precise enough, and we will enhance this part in the revision.
>
> -------------------------------------------------------------------------------------
>
> 3) *Experiments: Overall these experiments nicely demonstrate the methods introduced by the paper, but there are some missing details that should be added.*
>
> **Response:**
> We are sorry for missing some details of hyperparameters in the appendix, and we will add the following details in the revision of our paper.
>
> a)  We use 200 particles in MD3QN.
>
> b)  For kernel bandwidth, we use the bandwidths $B = [ 2^{-8}, 2^{-7}, 2^{-6}, \cdots, 2^8 ]$, as this is the best performing bandwidth according to our hyperparameter sweeps.
>
> c)  For learning rates, we follow IQN’s implementation and use lr=5e-5.
>
> d)  For Atari game/reward functions: we use the following criteria to select games and we have presented all the games we tried:
>
> - There are multiple sources of reward.
> - Different sources of reward can be recognized from scalar score signals.
>
> The way we compute the vectored reward is detailed in Appendix A.2. It is worth noticing that not all Atari games have multiple reward sources, and not all the rewards with different sources can be decomposed only from the scalar.
>
> For instance, in Gopher, the possible rewards for different events are 0 (for nothing), 20 (for filling the hole), 100 (for hitting the gopher). If r = 120, we can tell that it is composed of a reward of 20 and a reward of 100. However, in Seaquest, the agent will get the same reward for shooting the sharks and shooting the submarines. It is also hard to tell how many rewards are collected due to the leftover oxygen and how many rewards are collected due to saving the divers. This makes Sequest not meet the second criteria, and we omit Seaquest in our experiment. To conclude, we use the two aforementioned criteria to select the games that are of our interest.
>
> For Pong, the agent will get a reward = +1 for winning a round and a reward = -1 for losing a round. In fact, there are a lot of games with the same pattern of reward: +1 for winning a round and -1 for losing a round. However, we consider the sources of reward to be the same, so we only choose Pong as a representative game in this type of games.
>
> During the rebuttal, we go over all the Atari games and locate another two that meet the criteria and add them to the experiments. Finally, we choose: AirRaid, Asteroid, Gopher, MsPacman, Pong, UpNDown as the evaluation set. We further tune the hyperparameters and use kernel bandwidths $B=[2^{-8}, 2^{-7}, 2^{-6}, \cdots, 2^8]$.  The full results on Atari games with multiple random seeds are shown in [https://raw.githubusercontent.com/NeurIPS21-MD3QN/NeurIPS21-MD3QN-figures/main/Atari_HRA.png ].
>
> -------------------------------------------------------------------------------------
>
> 4) *"Distributional Reward Decomposition for Reinforcement Learning" (NeurIPS, 2019).*
>
> **Response:**
> We appreciate your comment! We will add this paper (Distributional Reward Decomposition for Reinforcement Learning) in our related work. This paper focuses on decomposing multiple types of rewards from one source reward, while in our paper, we are directly provided with multiple types of rewards as inputs, and model the joint distribution of different sources of rewards.
>
> -------------------------------------------------------------------------------------
>
> 5) *Proposition 1 & Typos*
>
> **Response:**
> Thanks for your advice! We have corrected all the typos. The metric of convergence in Prop.1 should be specified as $\overline{d}_p$.

---

### Official Review · Reviewer_fJbd · 2021-07-16

**Rating:** 6
**Confidence:** 2

**Summary:**

The paper proposes a new learning algorithm for modeling the distribution of multi-dimensional reward in the reinforcement learning (RL) setting. It proposes a bellman evaluation operator for the distribution of multi-dimensional reward and shows its optimality. Moreover, a MMD based metric is proposed to measure the difference between two distributions and this metric is fit for the typical RL learning settings. The new learning method is tested and performs the best on one designed maze environment and multiple Atari games.

**Limitations And Societal Impact:**

N/A.

**Main Review:**

Originality:
The idea of the learning distribution of multi-dimensional reward in RL is straightforward. But building a learning framework upon that is complex and significant.

Quality:

The learning framework is designed based on value-based reinforcement learning framework. The proof of the bellman operator is sound. Moreover, it uses different losses for proving and training. The MMD based training loss shows good performance in the experimental section.

Clarity:

The paper is well-written. And I do like the paper's structure.  It frames each main theorems and proposition, then explains the relation between them. Moreover, the authors also use many intuitions among them. Though the proof may not be fancy, however, the structure does make the understanding much easier.

Significance:

The problem to model the distribution of multi-dimensional reward is vital to push the performance of RL algorithms.

Questions:

1. I am wondering can you explain more about the kernel selection? Is any kernel that can make MMD a metric is a good kernel?
2. It would be better if you can compare MMD with some other distance metric and show the superiority of MMD.

Minor:
The $r_t$ of equation (2) should be bolded.

**Time Spent Reviewing:**

3

---

> ### Author Response · Authors · 2021-08-10
> **Response to Reviewer fJbd**
>
> Thanks for your thoughtful and constructive comments! Here we will answer the two questions accordingly.
>
> 1) *I am wondering can you explain more about the kernel selection? Is any kernel that can make MMD a metric is a good kernel?*
>
> **Response:**
> We are sorry that the kernel selection was not that clear in our original manuscript. We would like to apologize and explain it here.
>
> If the kernel is characteristic, then it can make MMD a metric. We choose the Gaussian kernel because it is a characteristic kernel, which makes the MMD a metric between distributions. This conclusion is proved by the original paper of MMD [1].
>
> As for the bandwidth selection, intuitively, we believe that the kernel bandwidths should cover a wide range. Here are the experiment results during the hyperparameter tuning, and we believe this can support this intuition. We tested 3 sets of kernel bandwidths: bandwidths $B_1 = [2^{-8}, 2^{-7}, 2^{-6}, \cdots, 2^8]$, bandwidths $B_2 = [0.01, 0.02, 0.04, 0.08, 0.16, 0.32, 0.48, 0.64, 0.80, 0.96, 2.0, 5.0, 10.0, 20.0, 50.0, 100.0]$, and bandwidths $B_3 = [1, 2, 3, 4, 5, 6, 7, 8, 9, 10]$, and perform the same experiment as in Section 5.1 and Figure 2 in our paper to test how MD3QN models the joint distribution under different bandwidth settings. The result can be found in  [ https://raw.githubusercontent.com/NeurIPS21-MD3QN/NeurIPS21-MD3QN-figures/main/bandwidth.png ]. For $B_1$, the MMD between prediction and ground truth distribution is 0.026, where for $B_2$, MMD = 0.036, and for $B_3$, MMD = 1.348. We found $B_1$ and $B_2$ can capture the joint distribution better than $B_3$, and $B_1$ is slightly better than $B_2$. Finally we use Bandwidth $B_3$ in our experiments.
>
> [1] A kernel two-sample test.
>
> -------------------------------------------------------------------------------------
>
> 2) *It would be better if you can compare MMD with some other distance metric and show the superiority of MMD*
>
> **Response:**
> We proposed an algorithm that is capable of learning the joint distribution of the return. To make our algorithm compatible with higher dimensions and also to avoid the curse of dimensionality, we use sample-based metrics here (to be specific, MMD) instead of measures used in C51 or QR-DQN that need some statistics that model the joint distribution directly. To be specific, we can get the unbiased estimation of MMD by samples, and we believe this is friendly to high dimensions. However, other distance measures, for instance, KL divergence (used in C51), will face the problems of dimension explosion. If we use KL divergence, the joint distribution may have extremely small probability in certain areas, and the KL divergence may converge to zero, which requires additional algorithm design to prevent from. Maybe we can try to design some other architectures and use other metrics in the future, but the need for additional algorithm design makes the comparison to other metric beyond the scope of this paper.

---

> > ### Comment · Reviewer_fJbd · 2021-09-15
> > **Post Rebuttal**
> >
> > Hi,
> >
> > Thanks for your detailed rebuttal. After the discussion with other reviewers. I think the paper needs to do a big revision to make the confusing part clear. So I downgrade the score to 6.

---

### Official Review · Reviewer_X5DH · 2021-07-19

**Rating:** 3
**Confidence:** 4

**Summary:**

The authors present MD3QN, which leverages the distributional approximation capacity of C51 and its likes as well as the reward multiplicity of some reinforcement learning tasks. The line or research centered around C51 has the advantage of learning an entire distribution of the reward (instead of just the expectation), which is often considered a better representation of the RL task at hand. Meanwhile, the line of research around HRA separately model the value function for each reward source to better the algorithm's sample efficiency. MD3QN combines the said advantages in a correlated multiple-reward setting where the agent learns the joint reward distribution for a better task representation and a better sample efficiency. The proposed algorithm is shown to outperform one previous work (HRA) in a number of Atari games, and it seems to learn the reward correlations from custom-designed mazes.

**Limitations And Societal Impact:**

The authors acknowledged that, despite MD3QN being a general framework, its actual application to goal-conditioned RL and successor states are left to future works. I believe the same kind of limitation acknowledgement is in order for i) testing with only four multiple-reward Atari games, ii) not clarifying a "common misunderstanding" in existing works despite claiming to have done so, and iii) the rather factitious nature of the maze experiments.

**Main Review:**

Originality:
I think the presentation could do a much better job at highlighting the novelty of the algorithm design. The current paper does not pinpoint the challenges encountered in developing MD3QN. I struggled to identify the unique contribution of MD3QN in the transition from Section 2 to Section 3, where MD3QN is illustrated almost like a mere combination of C51 and HRA when that is certainly not the case according to the introduction. The strength of MD3QN must then lie in capturing the correlated randomness in source-specific returns, but the experiments and the figures do not point to that strength. Figure 2 is especially obscure in that I wasn't convinced by its effectiveness or its message. How better does MD3QN do in identifying the correlations than do previous works? What quantitative measures could you utilize to gauge the performance in capturing the correlations? Also, why have the maze-multireward rewards been split into blue-orange and red-green pairs?

On a separate note, the paper claims multiple times that it "corrects the misunderstanding" of existing works, but I would like to raise a few points on this claim. The first mention of this claim is in Section 1, where "We correct a misunderstanding in existing works on distributional RL, and argue that we can use other metrics than the Wasserstein metric over the joint distribution as training losses to approximate the Bellman operator." The second and third mentions of the claim both appear in Section 3: "... misunderstanding that we must use the same metric for proving convergence and as training loss." Now, those two understandings are very different from each other, and the rest of the paper does not clarify which "misunderstanding" that the authors are "correcting". Is it that only the Wasserstein metric must be used? Or is it that the same metric must be used in convergence and in training loss? Furthermore, the first mention of the "misunderstanding in existing works" requires references. The second and third mentions of the "misunderstanding" do contain references to existing works, but as far as my understanding of Bellemare2017 and Nguyen2020 goes, those two papers in no way claim  that "only the Wasserstein metric must be used" or "the same metric must be used in convergence and in training loss". In fact, the MD3QN authors only list some other works that employ different metrics for showing convergence and in training loss. Consequently, I am lost as to which misunderstanding in which previous work the authors are referring to. If the "corrects the misunderstanding" statement is not addressed properly, it will put the submitted paper in a very bad light. You would be claiming to have corrected a misunderstanding that never even existed to begin with. Far worse than that would be to wrongly attribute that non-existent misunderstanding to a well-established prior art. It is very much needed that the authors cite exactly which sources make the said misunderstanding, but even that would not be enough to credit the MD3QN authors with "correcting a common misunderstanding" because many studies that the authors themselves have cited already use different metrics for convergence and in training loss. I am left puzzled as to why such a statement is needed in this paper after all.


Quality:
The paper was easy to read. There is considerable effort in founding a mathematical establishment before delving into the experiments, notations are consistent, as far as I can tell.
Some minor issues:
Section 2.1 is are sampled from the replay memory --> are sampled from the replay memory
Section 3.2 Crame loss --> Cramer loss
Section 5.1 The move has no effect --> The attempted move does not succeed (because, it's not that there is a move and then it has no effect. It's that there is no move in the first place.)


Clarity:
The sectioning feels a bit awkward. For example, what was meant to be the distinction between Section 2.1 and Section 3.1? At first, it looked like Section 2.1 is generally applicable to related research while Section 3.1 is specific to MD3QN, but that does not seem to be the case after continuing on to Section 3.2, where a concept (Bellman operator) much broader than this specific paper is introduced. The scopes of Sections 2 and 3 did not help in clarifying the notations, background, or the settings.

Again, referring to the "common misunderstanding" claim that the authors are making: authors write that they "clarify that the metric for proving convergence and the metric for training loss play different roles, and are not necessarily the same metric". No, they have not clarified that statement. They have cited a number of works that do not adhere to the "common misunderstanding", which then calls to question whether the said misunderstanding is indeed "common". It remains unclear i) who said that convergence metric and the training metric have to be the same, as well as ii) what mathematical tools the MD3QN authors have wielded to disprove the misconception.

The term "exclusive" is used without definition in Section 5.1. Is Figure 2 (a) supposed to show negative correlation?

Significance:
The key element of the paper is the correlatedness of the multiple rewards, and the two theorems do not show how the correlation plays a pivotal role in the design philosophy of MD3QN. Theorem 1 being the contraction mapping theorem for two joint distributions and Theorem 2 being the unbiasedness of the joint distribution estimator, neither of the two are, strictly speaking, contributions that are attributable to MD3QN, even if they show the "interesting" part when it comes to the correlation of the multiple rewards.

The experiments do not particularly outshine other submitted papers. For instance, the maze setting looks a bit artificial, and it would have been much more effective if accompanied with a real-world scenario, despite the fact that only some simplified analogues are tested.

Also, reward multiplicity poses some intriguing opportunities for rewards defined "outside the game". Intrinsic motivation papers geared towards better exploration or more diversity are examples of reward structure tuning. It would be enlightening to observe how MD3QN performs under those circumstances and whether the distributional RL advantage and/or the sample efficiency stand out in an ablation study.

The effectiveness of Figure 2 is unclear. The figures do not quantify how well MD3QN does, and they do not compare MD3QN to previous works. More importantly, why have the Atari Games been limited to just four games? For example, Atari Seaquest game grants points for destroying the sharks as well as some points for rescuing the divers. Why has this game been excluded?

**Time Spent Reviewing:**

3

---

> ### Author Response · Authors · 2021-08-10
> **Response to Reviewer X5DH**
>
> [ Post 1 / 2 ]
>
> Thanks for your thoughtful and constructive suggestions! We are eager to make some clarification and we have taken all the comments into consideration and summarized the responses as follows.
>
> 1) *I think the presentation could do a much better job at highlighting the novelty of the algorithm design.*
>
> **Response:**
> Thanks for your advice! We believe that maybe the novelty and challenges could be more clear in the original manuscript. Here we revise and formulate them as follows:
>
> a) Challenges in building the theoretical foundations:
> First of all, it is not trivial to extend the theory of C51 into multi-dimensional, although the proof is not (that) fancy. The Wasserstein metric has a closed-form in the 1-d situation, while in n-d, we can only use the coupling form in the proof. Therefore, we use new proof methods in Lemma 3 and Lemma 4, which both use the coupling form of Wasserstein distance. In Lemma 3, we prove that if the Wasserstein distance of two joint distributions is $C$, then after applying the same linear transform to these two distributions with a scale factor of $\gamma$, the Wasserstein distance is at most $\gamma C$. In Lemma 4, we prove that the Wasserstein distance of two joint distributions under conditional random variable $A$ is no greater than the maximum Wasserstein distance overall specific conditions ($A=a$). Both the proof of Lemma 3 and Lemma 4 are not proposed in the existing works. By the results of Lemma 3 and Lemma 4, we are able to prove the contraction results in Theorem 1.
> To the best of our knowledge, we are the first to give rigorous proof for the contraction of the multidimensional Bellman evaluation operator and the contraction of the expectation summation of the multidimensional Bellman optimality operator, although it seems to be intuitive.
>
> b) Joint distribution is very useful in downstream tasks:
> As you mentioned, the strength of our algorithm lies in capturing the joint distribution, and we believe this is very useful in many real-world scenarios. In our paper, we use a toy case and Atari games to show the validity of our algorithm.
>
> As you mentioned in (8), we agree that maybe those experiments are not explicit enough to elaborate the strength of our algorithm. To better highlight our novelty and significance,  we add another experiment to show the necessity of modeling the joint distribution, and explain why only modeling marginal distributions (as a trivial extension of MMDQN to high dimensions) may fail in some settings (e.g., in environments with multiple correlated constraints) from theory and experiment results. For detailed information, please refer to (8).
>
> -------------------------------------------------------------------------------------
>
> 2) *Figure 2 is especially obscure in that I wasn't convinced by its effectiveness or its message. How better does MD3QN do in identifying the correlations than do previous works? What quantitative measures could you utilize to gauge the performance in capturing the correlations? Also, why have the maze-multireward rewards been split into blue-orange and red-green pairs?*
>
> **Response:**
> Thanks for your advice! We also agree that it would be better to explain the experiment results more clearly.
>
> Firstly, the experiment results corresponding to figure 2 show that our algorithm is capable of capturing the joint distribution. We argue that none of our baselines can do the same thing, because they either learn the one-dimensional return or learn the scalar multidimensional returns. As for the quantitative measure, we believe this is a good idea, and we carry out a quantitative experiment to show the correlations are indeed learned through the training process. We perform the same experiment as in Figure 2 in our paper, which tests how MD3QN models the joint distribution, and record MMD between the joint distribution prediction and the ground truth distribution along the training process. We show the training curve of three environments, "maze-multireward", "maze-exclusive" and "maze-identical" with the Y-axis showing the MMD metric. This result can be found in [ https://raw.githubusercontent.com/NeurIPS21-MD3QN/NeurIPS21-MD3QN-figures/main/Fig2Curve-maze0.png ,  https://raw.githubusercontent.com/NeurIPS21-MD3QN/NeurIPS21-MD3QN-figures/main/Fig2Curve-maze1.png ,  https://raw.githubusercontent.com/NeurIPS21-MD3QN/NeurIPS21-MD3QN-figures/main/Fig2Curve-maze2.png ]. We add some visualizations in the curve to intuitively show the relation between the value of MMD and the samples.
>
> Our algorithm models the 4-dimensional joint distribution and the blue-orange & green-red combinations are only picked for the convenience of visualization, and they are not special. We plot the results for all combinations in [ https://raw.githubusercontent.com/NeurIPS21-MD3QN/NeurIPS21-MD3QN-figures/main/combination.png ].
>
> -------------------------------------------------------------------------------------
>
> 3) *Concerns on our claim about the "common misunderstanding"*
>
> **Response:**
> We are sorry that this point was not clear in our original manuscript, and we apologize for all the confusion.
>
> First of all, we agree that the expression in the original manuscript is not rigorous enough. We would like to restate our claim: (1) we can use a metric A to prove that the (projected) distributional Bellman operator is a contraction, and (2) we can use another distribution measure B as training loss, as long as global minimization of this loss lead to the same result as the application of the (projected) Bellman operator.
>
> To be specific, for the distributional Bellman operator, the metric A used for proof and measure B used as loss are not necessarily the same. In our case, we use MMD loss as B, and the above claim holds when the number of samples goes to infinity. For the projected distributional Bellman operator, there exists another metric used for projection, and this is not necessarily the same as B, too.
>
> Since (projected) Bellman operator with metric A is a contraction, and global minimization of loss B has the same result (projected) Bellman operator, it is natural to conclude that it is okay to use B as training loss with the theoretical guarantee proved by A not broken.
>
> As for the “common misunderstanding”, we believe that maybe it’s not that appropriate to use this expression, and we apologize for all the confusion. However, we believe that there do exist some expressions in the existing work that are kind of misleading.
>
> In [1], they raise “an open question about the sufficient condition for contraction”, suggesting that they think that they need to prove contraction under MMD metric since they use MMD as their loss. However, the convergence result is already built with the contraction result under the Wasserstein metric in [2]. Based on the aforementioned works, we believe that there does exist some misleading information in existing work.
>
> We adopt this claim, because we use wasserstein metric to prove convergence of two Bellman operators, and use MMD metric as training loss. Therefore we believe it’s better to clarify whether it's okay to use different metrics as proof of convergence and training loss.
>
> [1] Distributional Reinforcement Learning via Moment Matching.
>
> [2] A Distributional Perspective on Reinforcement Learning.
>
> -------------------------------------------------------------------------------------
>
> 4) *Typos.*
>
> **Response:**
> Thanks for your advice! We have corrected the typos and used the more precise expression as you suggested.
>
> -------------------------------------------------------------------------------------
>
> 5) *The sectioning feels a bit awkward.*
>
> **Response:**
> Thanks for your advice! We agree that it will be more clear to introduce the normal Bellman operator in Section 2. However, in Section 3.2, we introduce a specific Bellman operator (the joint distributional Bellman operator) that is closely related to our paper. It is worth noticing that this is not the broad Bellman operator. To the best of our knowledge, we are the first to propose this specific Bellman operator. We build the theoretical foundations for our algorithm based on this specific Bellman operator and our algorithm also approximately applies this operator.
>
> -------------------------------------------------------------------------------------
>
> 6) *The term "exclusive" is used without definition in Section 5.1. Is Figure 2 (a) supposed to show negative correlation?*
>
> **Response:**
> We apologize for the ambiguity. We have a brief definition in general (l.270 - l.271): Once the agent obtains a reward from one side, it cannot obtain the reward from the other side.
> Here is a more detailed explanation: the yellow wall only allows the agent to move from left to right and from up to down, but will block the moves from the opposite. What’s more, the attempted move to a previous position will not succeed (l.265). This means that the agent can either collect the red one or the green one at each connected area. After the collection, the only move that will succeed is to go through the yellow wall, and it can never come back to collect the other one. Therefore, Figure 2(a) is supposed to be negatively correlated.

---

> > ### Author Response · Authors · 2021-08-10
> > **Response to Reviewer X5DH 2**
> >
> > [ Post 2 / 2 ]
> >
> > 7) *The key element of the paper is the correlatedness of the multiple rewards, and the two theorems do not show how the correlation plays a pivotal role in the design philosophy of MD3QN.*
> >
> > **Response:**
> > We would like to elaborate why the two theorems are related to the correlatedness of the multiple rewards here:
> > Theorem 1, 2 shows the contraction of the joint distributional Bellman evaluation operator and the optimality operator accordingly. To the best of our knowledge, we are the first to provide rigorous proof. Those two theorems serve as the theoretical guarantees of correctness for our proposed algorithm and even future works in high dimensions. Given the two theorems, it can be seen that by applying the Bellman operator, we can get the joint distribution that encodes the correlation information. From the joint distribution of random variables, we can precisely tell if they are positively or negatively correlated. Therefore, our two theorems support modeling the correlation of different rewards.
> >
> > -------------------------------------------------------------------------------------
> >
> > 8) *The experiments do not particularly outshine other submitted papers. For instance, the maze setting looks a bit artificial, and it would have been much more effective if accompanied with a real-world scenario, despite the fact that only some simplified analogues are tested.*
> >
> > **Response:**
> > To further illustrate the necessity of learning a full joint distribution, we add another experiment during the rebuttal, and explain why only modeling marginal distributions may fail in some settings (e.g., in environments with multiple correlated constraints)  from theory and experiment results.
> >
> > There are a number of real-world scenarios where multiple constraints should be met simultaneously. For instance, in autonomous driving, we need to balance the safety (distance from other cars), the speed, the comfort (the acceleration, etc.), and many other factors to make sure the car functions normally. Those statistics can be viewed as sub-rewards in our settings. For instance, we would like to make the agent be aware of the joint distributions of distance from other cars, the speed, and add a constraint on this distribution (e.g., d > 1m, v < 50km/h).
> >
> > From the theoretical perspective, only the algorithm which is capable of modeling the joint distribution can find this optimal solution. If the algorithm can only model the marginal distributions, we can correctly compute the probability of meeting multiple constraints only if they are independent. However, this is not common in real-world scenarios.
> >
> > We use the same Maze environment as in Figure 1 of our paper and modify the layout as a simplified analogues (see [ https://raw.githubusercontent.com/NeurIPS21-MD3QN/NeurIPS21-MD3QN-figures/main/constraint_maze.png ]). We set multiple constraints on the total return (in our experiment, we have three constraints: for the initial state, total red return > 0.6, total green return > 0.6, and total blue return > 0.6, where the agent gets a reward = 1.0 of the specific color after collecting a specific colored block), and the goal is to find a policy which has the highest probability to satisfy all constraints.
> >
> > We use the joint distribution by MD3QN to achieve this: specifically, given the modeled joint distribution $\textbf{Z}(s,a)$, the agent can compute the probability to satisfy all the constraints in the joint distribution and take action by $\arg \max_a \mathbb{P}(\textbf{Z}(s,a) \text{ satisfy all three constraints})$. We also modify the joint distributional Bellman optimality operator to maximize the probability to satisfy all constraints.
> >
> > For baseline methods, we extend the MMDQN by multiple heads $Z_{1:N}$ to reflect $N$ sources of rewards. This method is different from our algorithm in that only marginal distribution of each dimension is learned. We test two ways to approximately compute the probability to satisfy all constraints by marginal distributions: the Marginal_SUM baseline maximizes $\sum_{i=1}^N \mathbb{P}(Z_i(s,a) \text{ satisfy i-th constraint})$, the sum of probabilities for each reward to satisfy constraint, while the Marginal_PROD baseline maximizes $\Pi_{i=1}^N \mathbb{P}(Z_i(s,a) \text{ satisfy i-th constraint})$, the product of probabilities for each reward to satisfy constraint.
> >
> > The results can be found here [ https://raw.githubusercontent.com/NeurIPS21-MD3QN/NeurIPS21-MD3QN-figures/main/multiple_constraint.png ]. It can be seen that maximizing the probability based on joint distribution can significantly outperform two baseline methods which both use marginal distribution information.
> >
> > -------------------------------------------------------------------------------------
> >
> > 9) *Why have the Atari Games been limited to just four games?*
> >
> > **Response:**
> > We are sorry that this point was not clear in our original manuscript, and we will add the following details in the revision of our paper.
> >
> > On the experiment to evaluate the performance of MD3QN, we use two criteria for selecting the Atari games:
> > - There are multiple sources of reward.
> > - Different sources of reward can be recognized from scalar score signals.
> > The way we compute the vectored reward is detailed in Appendix A.2. It is worth noticing that not all Atari games have multiple reward sources, and not all the rewards with different sources can be decomposed only from the scalar.
> >
> > For instance, in Gopher, the possible rewards for different events are 0 (for nothing), 20 (for filling the hole), 100 (for hitting the gopher). If r = 120, we can tell that it is composed of a reward of 20 and a reward of 100. However, in Seaquest, the agent will get the same reward for shooting the sharks and shooting the submarines. It is also hard to tell how many rewards are collected due to the leftover oxygen and how many rewards are collected due to saving the divers. This makes Sequest not meet the second criteria, and we omit Seaquest in our experiment. To conclude, we use the two aforementioned criteria to select the games that are of our interest.
> >
> > For Pong, the agent will get a reward = +1 for winning a round and a reward = -1 for losing a round. In fact, there are a lot of games with the same pattern of reward: +1 for winning a round and -1 for losing a round. However, we consider the sources of reward to be the same, so we only choose Pong as a representative game in this type of games.
> >
> > During the rebuttal, we go over all the Atari games and locate another two that meet the criteria and add them to the experiments. Finally, we choose AirRaid, Asteroid, Gopher, MsPacman, Pong, UpNDown as the evaluation set. We further tune the hyperparameters and use kernel bandwidths B=$[2^{-8}, 2^{-7}, ..., 2^8]$. The full results on Atari games with multiple random seeds are shown in [ https://raw.githubusercontent.com/NeurIPS21-MD3QN/NeurIPS21-MD3QN-figures/main/Atari_HRA.png ].

---

### Official Review · Reviewer_A2nn · 2021-08-01

**Rating:** 5
**Confidence:** 3

**Summary:**

The paper suggests exploiting the awareness and available identification of multiple reward sources to factorise a distributional value function and exploit this explicit structure to learn and exploit the correlation structure of distinct reward sources. The overall architecture predicts representative samples that describe the joint (total) value distribution. The paper presents experimental results that confirm efficacy of the approach. The paper appears to be correct overall, although there are some technical issues that mar the overall impression.


**Limitations And Societal Impact:**

Work is sufficiently distant from real world applications to avoid social impact.

**Main Review:**

First noticeable issue with the paper is Algorithm-1. Authors suggest that it is supposed to calculate the gradient of the MMD loss. However, it actually calculates an estimate of the MMD loss itself. This is in a sharp contrast to the preceding paragraph that speaks of the necessary simplifications. The appendix is of no help in this situation. Authors do not describe how the loss estimate will translate into it's gradient estimate. The text of the algorithm is also festered by repeated use of symbols at distinct semantic locations ("a prime" serves too many roles here, so is Z_{1:M}). It is difficult to imagine how this pseudo-code was translated into the actual code to function correctly. Notwithstanding the practical success of the code, this does not bode well for the alignment of the theory and practice of this paper.

It is also not clear why borrowing a few factorisation tricks from the MARL literature, where the correlation of independent executions is implicitly learned, can not be of use here. QR-MIX and MMD-MIX would be among the more recent structures that can be successfully deployed here. Authors cannot avoid a justification of not using these architectures (at least as a baseline) in their domain.

Authors also miss more important references that bear key similarities to their approach:
"Distributional Multivariate Policy Evaluation and Exploration with the Bellman GAN" by Freirich et al
"Sample-based distributional policy gradient" by Singh, Lee and Chen
It would be interesting if authors could compare their work to the above two (at least in the rebuttal).

Peeves and typos:

The paper is also overly reliant on the Appendix. Though the claims to appear to be correct, there are intuitions that may be gathered here, that do not appear in the main text.

Line 169 "Crame_r"
Line 170 "Operator"
Line 277, what is the exact nature of an "iteration" here?



**Time Spent Reviewing:**

~5h (with extra noise)

---

> ### Author Response · Authors · 2021-08-10
> **Response to Reviewer A2nn**
>
> Thank you for the thoughtful and constructive suggestions! We have taken all the comments into consideration and summarized the responses as follows:
>
> 1) *First noticeable issue with the paper is Algorithm-1.*
>
> **Response:**
>
> Thanks for your advice! We agree that the original manuscript might not be so clear. We would like to revise our expressions here:
>
> Algorithm 1 actually computes the gradient of MMD^2, instead of MMD^2 itself. The $\nabla_\theta$ in line 6 of Algorithm 1 denotes the gradient of MMD^2 with respect to the parameter $\theta$, which is computed by backpropagation. Since the functions in Algorithm 1 to compute MMD^2, $\boldsymbol{\mu}(s,a;\theta)$ and $k(\cdot, \cdot)$, are differentiable, we can compute $\nabla_\theta \text{MMD}^2$ by existing automatic differentiation frameworks, like TensorFlow.
>
> About notation overload: we really appreciate your comments, and we will replace $a’$ to $a$ in the argmax operator, and replace $Z_{1:M}$ to $Z^{1:M}$ to avoid conflicted notations.
>
> -------------------------------------------------------------------------------------
>
> 2) *It is also not clear why borrowing a few factorisation tricks from the MARL literature, where the correlation of independent executions is implicitly learned, can not be of use here.*
>
> **Response:**
> First of all, we would like to clarify that, instead of taking only one-dimensional rewards from the environment and learning to factorize this scalar reward to multiple sources (We think there might be some misunderstanding here? Because “The paper suggests exploiting the awareness and available identification of multiple reward sources to factorise a distributional value function...”), our algorithm directly takes multiple rewards from the environment as input. Namely, at each timestep, the environment returns a vector of reward (l. 79-80 for definition).
> In MARL, the agent learns to decompose a joint Q into individual Qs and we assume this is what the “factorisation tricks” refer to.
> However, we believe that our work is different from MARL in terms of problem settings (we don’t need the factorisation step). So, maybe it is not necessary to use or compare with QR-MIX / MMD-MIX here.
>
> -------------------------------------------------------------------------------------
>
> 3) *It would be interesting if authors could compare their work to the above two (at least in the rebuttal).*
>
> **Response:**
> Thanks for your advice. We will add the following discussion to the revision of our paper. As for the paper “Distributional Multivariate Policy Evaluation and Exploration with the Bellman GAN”, it only supports policy evaluation, while our algorithm can handle both policy evaluation and optimization. As for “Sample-based distributional policy gradient”, it is a policy gradient-based method for one-dimensional return, while our method is a value-based method for multidimensional return.
>
> -------------------------------------------------------------------------------------
>
> 4) *Peeves and Typos.*
>
> **Response:**
> We really appreciate your advice! We have corrected the typos and added some intuitions in the main text.

---

### Author Response · Authors · 2021-08-19
**A kind reminder**

Dear reviewers,

Thank you for your time and efforts in reviewing our paper.

This is a kind reminder that we are one week into the discussion period. We believe that we have sincerely and successfully addressed your concerns and questions in your reviews, with the added supporting experimental results.

If you have any further questions or concerns after reading our comments, please do not hesitate to let us know.

Authors

---

### Decision · Program_Chairs · 2021-09-28

**Decision:**

Accept (Poster)

**Comment:**

The reviewers generally like the fundamental ideas of this paper.  However, there was a lot of confusion regarding the notation and the supposedly common misunderstandings of the community. The reviewers also raised some concerns about the experiments.  While the rebuttal clarified many concerns, the revisions required for the paper to be accepted are significant and therefore another round of reviews will be needed.  Hence, the authors are encouraged to revise the paper and to resubmit it at another venue.

**Consistency Experiment:**

NeurIPS has a long history of experimentation. In 2014, NeurIPS ran an experiment in which 10% of submissions were reviewed by two independent committees to quantify the randomness in the review process. This year, we repeated a variant of this experiment to see how the quality of the review process has changed over time.  This paper was part of the experiment and was therefore assigned to two committees (consisting of reviewers, an Area Chair, and a Senior Area Chair) that reached independent decisions.  If both committees made the same recommendation, this recommendation was followed. If a single committee recommended acceptance, the paper was accepted (with the exception of a few cases in which the other committee identified what we considered a fatal flaw, e.g., an error in a key result).

This copy’s committee reached the following decision: **Reject**

The other committee assigned to the paper recommended **Accept (Poster)**.  You can find the other set of reviews, along with any follow up discussion with the authors here:
https://openreview.net/forum?id=u7oKU1iXTa9